# A MARGIN-BASED REPLACEMENT FOR CROSS-ENTROPY LOSS

## ABSTRACT

Cross-entropy (CE) loss is the de-facto standard for training deep neural networks (DNNs) to perform classification. Here, we propose an alternative loss, high error margin (HEM), that is more effective than CE across a range of image-based tasks: unknown class rejection, adversarial robustness, learning with imbalanced data, continual learning, and semantic segmentation (a pixel-wise classification task). HEM loss is evaluated extensively using a wide range of DNN architectures and benchmark datasets. Despite all the experimental settings, such as the training hyper-parameters, being chosen for CE loss, HEM is inferior to CE only in terms of clean and corrupt image classification with balanced training data, and this difference is small. We also compare HEM to specialised losses that have previously been proposed to improve performance for specific vision tasks. LogitNorm, a loss achieving state-of-the-art performance on unknown class rejection, produces similar performance to HEM for this task, but is much poorer for continual learning and semantic segmentation. Logit-adjusted loss, designed for imbalanced data, has superior results to HEM for that task, but performs worse on unknown class rejection and semantic segmentation. DICE, a popular loss for semantic segmentation, is inferior to HEM loss on all tasks, including semantic segmentation. Thus, HEM often out-performs specialised losses, and in contrast to them, is a general-purpose replacement for CE loss.

## 1 INTRODUCTION

Deep neural networks (DNNs) are generally trained using variants of stochastic gradient descent. These optimisers require the loss function to have a gradient. This means that it is not possible to maximise the classification accuracy directly as this function is piece-wise constant, and therefore, does not define a usable gradient. As a result, it is necessary to use a *surrogate* loss function that has a gradient, but still encourages few classification errors. The design of the loss function is important as different losses will lead to different training speeds and cause convergence to different parameters. While many possible surrogate loss functions have been proposed (Wang et al., 2022; Terven et al., 2025), cross-entropy (CE) loss is by far the most common choice for classification tasks.

In some specific domains better performance can be obtained by using other losses. For example, alternative losses and regularisation terms have been proposed to improve robustness to adversarial attack (Cui et al., 2024; Mao et al., 2019; Tack et al., 2022; Zhang et al., 2019; Kannan et al., 2018; Kanai et al., 2023; Awasthi et al., 2023; Yu & Xu, 2023; Panum et al., 2021; Pang et al., 2020). In the domain of open-set recognition, where the aim is to better detect and reject images from "unknown" classes, alternative losses (such as contrastive losses) have been employed together with architectural modifications to improve performance beyond that achieved by CE (Zhu et al., 2023; Ming et al., 2023). Alternatively, LogitNorm loss (Wei et al., 2022), can be employed to improve unknown class rejection without the need for modifications to the network architecture or training procedure. To deal with situations where the training data contains drastically different numbers of samples for different classes ("class imbalance"), state-of-the-art approaches use a logit-adjusted loss which weights minority classes more heavily (Menon et al., 2021; Ren et al., 2020; Cui et al., 2019). For semantic segmentation, where the aim is to assign a class label to each image pixel, the two largest groups of losses centre around CE and its variants, and DICE and its variants (Ma et al., 2021; Azad et al., 2023). DICE loss (Milletari et al., 2016) is designed to be particularly effective when there is class imbalance, a common situation in segmentation tasks. While these specialised losses can

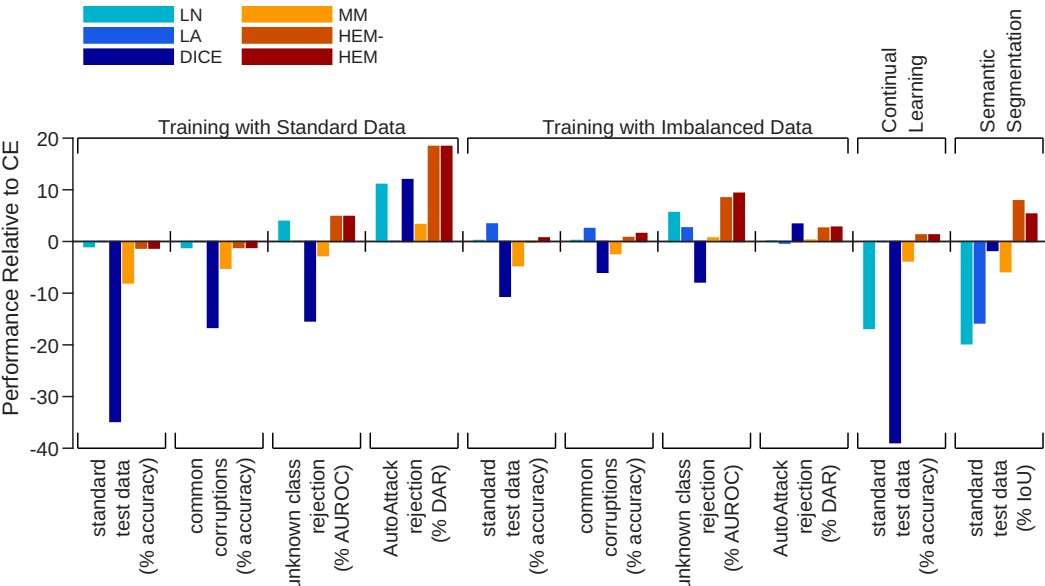

Figure 1: Summary results for all the different tasks considered, comparing the average performance of cross-entropy (CE) loss to each of the alternative losses that have been evaluated: LogitNorm (LN), Logit-adjusted (LA), DICE, multi-class margin (MM), high error margin with shared margin (HEM-), and high error margin with adjusted margins (HEM). Results are averaged using the arithmetic mean over all other factors that were varied in the experiments. Specifically, for all experiments on "Training with Standard data" (the first four segments of the figure), each bar is an average of 71 experiments (5 data-sets x 3 network architectures x 5 repeats, except for one combination of data-set and architecture where only one trial was performed). For all experiments on "Training with Imbalance Data" (the fifth to eighth segments) each bar is an average of 75 experiments (5 data-sets x 3 network architectures x 5 repeats). For the experiments on continual learning (the ninth segment), each bar is an average of 80 experiments (4 data-sets x 4 continual learning techniques x 5 repeats). For experiments on semantic segmentation (the tenth segment), each bar is an average of 76 experiments (3 data-sets x 4 network architectures x 5 repeats + 1 data-set x 4 network architectures x 4 repeats). For all the evaluation metrics used, higher values indicate better performance. The relative performance is calculated by subtracting the performance produced by CE loss from the corresponding metric for each of the other losses. Hence, positive values indicate average performance better than that of CE loss. Note that the results for LA are equal to those of CE, and the results for HEM are equal to those of HEM- when the training data is balanced: *i.e.*, when using standard training data (results in segments 1 to 4 of the figure) and when performing continual learning (segment 9).

outperform CE on the specific tasks for which they were developed, they tend to perform poorly outside of their specialised domain, as is confirmed by our results which are summarised in Fig. 1.

The fact that specialised losses out-perform CE on certain tasks motivates the search for a better classification loss function, that performs well on a range of tasks. For simpler statistical models, margin based losses (Crammer & Singer, 2002) are known to be general-purpose and show better generalisation behaviour than CE loss. We hypothesised that similar advantages could be achieved for DNNs by using a margin based loss. Particularly, we expected a margin-based loss to train networks that were less susceptible to making over-confident predictions, and hence, that would be better able to distinguish known from unknown classes. Furthermore, a margin-based loss should be less prone to over-write previously learnt weights, which could reduce catastrophic forgetting in continuous learning and over-writing weights necessary to classify minority classes when training with imbalanced data. Hence, a margin-based loss is a promising candidate for a general-purpose classification loss function. However, the existing multi-class margin-based loss (MM) results in performance that is frequently much worse than that achieved with CE loss (Fig. 1).

Table 1: Example outputs (logits), $\mathbf{y}$, from the last layer of an imaginary neural network being trained to perform a 4-way classification task, and the corresponding losses associated with each of these predictions when the first output represents the correct class. Hyper-parameters for each loss were set to $\tau = 1$ for cross-entropy (CE) loss, $\tau = 0.04$ for LogitNorm (LN) loss, and $\mu = 0.5$ for multi-class margin (MM) and high error margin (HEM).

| $\mathbf{y}$ | CE | LN | MM | HEM |
|---|---|---|---|---|
| $[1.0 \quad -1.0 \quad -1.0 \quad -1.0]$ | 0.34 | 0.00 | 0.00 | 0.00 |
| $[0.6 \quad 0.1 \quad 0.1 \quad 0.1]$ | 1.04 | 0.00 | 0.00 | 0.00 |
| $[0.6 \quad 0.3 \quad 0.0 \quad -0.1]$ | 1.02 | 0.00 | 0.11 | 0.45 |
| $[0.6 \quad 0.5 \quad 0.0 \quad -0.7]$ | 1.00 | 0.09 | 0.16 | 0.63 |
| $[0.6 \quad 0.7 \quad 0.0 \quad -3.5]$ | 0.98 | 1.10 | 0.19 | 0.77 |
| $[0.0 \quad 1.0 \quad 0.0 \quad 0.0]$ | 1.74 | 25.00 | 0.66 | 0.88 |

Here, we propose high error margin (HEM) loss, a new margin-based loss function that can be used as a general-purpose replacement for CE loss. To motivate our new loss we first describe issues with CE loss (Section 2.1) and MM loss (Section 2.2). Section 3 describes HEM loss which fixes the shortcomings of the existing losses that we have identified in our analysis in Section 2. Finally, we present extensive evaluations performed using nineteen different neural network architectures (ranging in size from LeNet to Vit-B/16) trained on many different data-sets (ranging in size from MNIST to ImageNet1k). We find that HEM is competitive with or out-performs CE loss across a range of classification tasks (Section 4). Full details of CE and all the other existing loss functions that we consider are given in Appendix A.

## 2 ANALYSIS AND MOTIVATION

### 2.1 ISSUES WITH CE LOSS

Even when a classifier produces the correct classification with high confidence, the CE loss is far from zero (Table 1, row 1). Consequently the gradients will be non-zero and each presentation of a correctly classified training sample will cause the weights to be modified so that the outputs become ever more extreme (highly positive for the logit representing the correct class and highly negative for the logits representing the incorrect classes). CE loss therefore encourages a DNN to map every training exemplar to an output where confidence in the predicted classification is extremely high. It is to be expected that such a DNN will produce high confidence for all samples, including ones not seen during training. It is unsurprising, therefore, that CE-trained DNNs have issues using prediction confidence to distinguish known from unknown classes.

CE loss continuing to update the weights even when the correct classification has been learnt successfully, will cause weights to be over-written. This behaviour may underlie some of the issues when CE loss is used for continual learning and learning with imbalanced training data. For good performance in these tasks it is necessary to maintain weights that represent classes learnt earlier or that have few training samples. Even after the classifier has achieved perfect performance, CE loss will update the weights further causing some forgetting of the previously learnt classes or the minority classes. This is consistent with the observation that models learn fewer features relevant to the minority classes than the majority classes (Dablain et al., 2024).

Another issue with CE loss is that it can be lowered by reducing the logit for an already clearly rejected class without increasing the difference to the closest competitor class. As a result, CE loss can behave quite counter-intuitively: decreasing even though the prediction is becoming poorer (Table 1, second to third rows). Most concerningly, CE loss can even be lower for incorrect classification (Table 1, penultimate row), than for correct classification.

LogitNorm (LN), multi-class margin (MM) and the proposed high error margin (HEM) losses, do not suffer from these issues. These losses produce a loss of zero for samples where the output of the neuron representing the true class is sufficiently larger than the outputs of other neurons (Table 1 top rows) and non-zero losses only for predictions that are worse at distinguishing the true class from the

alternatives (Table 1 bottom rows). In the case of MM and HEM, this is due to these losses using a margin. In the case of LN loss this is due to it behaving like a margin loss (Appendix A.2.1).

## 2.2 ISSUES WITH MM LOSS

The analysis in Section 2.1 suggests that a margin-based loss should have advantages over CE loss. A margin loss, including our proposed High Error Margin (HEM) loss, defines an error, $e_i$, associated with each logit, $y_i$, as follows:

$$e_i = \begin{cases} \max(0, y_i - y_l + \mu_i) & \text{if } i \neq l \\ 0 & \text{if } i = l \end{cases} \tag{1}$$

where $\mu_i$ is the margin (a non-negative hyper-parameter) for logit $i$, and $l$ is the index corresponding to the correct class (*i.e.*, the ground-truth class label). The error is zero when the output from the neuron representing the correct class exceeds the outputs of the other logit by at least the margin.

MM loss, the existing margin-based loss (see Appendix A.5 for full details), produces classifiers that have significantly lower accuracy on the standard test data compared to equivalent CE-trained classifiers (as shown in Fig. 1). We put this down to the method used to combine the errors. In MM loss this is achieved by averaging (or summing) the errors (across logits and samples in the batch). We believe this method of combining the error causes the gradient magnitude to change too much over the course of training. The loss is much higher at the start of training than near its end, because most errors become zero. This effect either prevents the suppression of the last non-zero errors towards the end of training or leads to instability at the start. The severity of this problem is increased for tasks with more classes.

# 3 HIGH ERROR MARGIN LOSS

We propose to solve the training issues of MM loss (Section 2.2) by combining the errors (Eq. (1)) in a more adaptive way to reduce the change in gradient magnitude during training. For each sample, all error values below the mean are set to zero, and the mean of above-zero values is calculated:

$$\mathcal{L}_{HEM} = \frac{\sum_{i=1}^{n} \left( \mathbb{1}[e_i \geq \frac{1}{n} \sum e_j] \times e_i \right)}{\sum_{i=1}^{n} \mathbb{1}[e_i \geq \frac{1}{n} \sum e_j]} \tag{2}$$

where $n$ is the number of classes, and $\mathbb{1}[\cdot]$ is the indicator function, which equals 1 if the argument is true and 0 otherwise. We call this loss the high error margin (HEM) loss, as it takes the average only of high errors. Losses for different samples in the batch are also combined by finding the mean of above-zero values. Note that the computation of the mean error, used by the indicator function in Eq. (2), is detached from the computational graph so that it does not affect the gradients.

At the start of learning, when a large number of logits produce errors (especially when $n$ is large), the mean error for each sample will be significant and, by only considering those losses above the mean, HEM concentrates on reducing the largest errors. Later in learning, there will be many zero errors and as a result, the mean error will be small (likely smaller than the few non-zero errors that remain). Hence, at this stage in training thresholding the errors by the mean will have little effect. However, by taking the mean of only the above-zero values, the loss will remain large even when there are few non-zero errors in each sample, and/or few incorrectly classified samples in a batch. As a result, HEM loss concentrates on the logits that produce the highest errors throughout learning. The effectiveness of the proposed method of combining errors, compared to that used in MM loss, was confirmed in an ablation study presented in Section 4.5.

We present results for two variants of the proposed loss:

**High Error Margin with adjusted margins (HEM)** which uses per-logit margins as defined in Eq. (1). Each margin was set to be inversely proportional to the number of training samples associated with that class. Specifically, $\mu_i = \sqrt{M/(ns_i)}$, and $s_i$ is the number of samples in class $i$. The separate, class specific, margins help deal with class imbalance.

**High Error Margin with shared margin (HEM-)** which (like MM loss) sets all margins to the same value (*i.e.*, $\mu_i = \mu \; \forall i$). The shared margin was made equal to $\sqrt{M/\sum_{i=1}^{n} s_i}$. HEM-

is an ablated version of the proposed loss that we use to provide a fairer comparison with CE and MM losses which do not employ mechanisms to deal with class imbalance.

Both versions employ a single hyper-parameter, $M$. Based on preliminary experiments (see Appendix D.1) $M$ was set to a value of 2000 for all other experiments described in this paper. Note that when the training data contains the same number of samples per class, all the margins used by HEM are equal, and HEM is identical to HEM-.

## 4 RESULTS

### 4.1 LEARNING WITH STANDARD DATA-SETS

We compared the performance of networks trained with our proposed loss, HEM, to the performance of identical networks trained with the alternative losses described in Appendix A. We have sought to produce a fair and representative evaluation of the different losses by using many different tasks, data-sets, and network architectures. Performance was tested using a number of different metrics to assess accuracy, generalisation, and robustness. For all metrics larger values correspond to better performance. Full details of the tasks, data-sets, evaluation metrics, DNN architectures and training set-ups are provided in Appendix B.

The tasks were chosen to include essential and important applications in computer vision (image classification and semantic segmentation), and tasks where we expected our margin loss to perform well, due to it not learning to make predictions with very high confidence (unknown class rejection) and stopping weight updates once adequate performance is achieved (continual learning and learning with imbalanced data).

For each experimental condition (combination of data-set, network architecture, and loss function) a network was trained and evaluated multiple times (typically five), each time with a different random weight initialisation and random presentation order of training samples. In the main text, summary results are presented by showing the average performance relative to CE loss. Detailed results showing absolute performance together with error-bars are reported in Appendix C.

Each experiment was performed using a single NVIDIA Tesla V100 GPU with 16GB of memory, except for experiments with ImageNet1k which were executed in parallel on four such GPUs. Performance differences can be interpreted independently of computational cost because the time taken to compute any of the losses is negligible compared to the overall execution time.

Performance on standard image classification tasks was assessed using five benchmark data-sets: MNIST, CIFAR10, CIFAR100, TinyImageNet and ImageNet1k. For each training data-set experiments were performed using three different neural network architectures (see Table 3 in Appendix B.1.3). Performance was evaluated in terms of the following criteria: accuracy on standard test data, accuracy on common corruptions test data, ability to identify and reject samples from unknown classes, and the proportion of adversarial samples correctly classified or rejected (details in Appendix B.1).

### 4.1.1 PERFORMANCE ON STANDARD TEST DATA AND COMMON CORRUPTIONS DATA

Networks trained using CE loss and HEM loss (here, because the training data is balanced, HEM- = HEM) have comparable performance on classifying the standard test data and the common corruptions data (Figs. 1 and 2, $1^{st}$ and $2^{nd}$ segments), although CE loss has a small advantage. On average this advantage for clean accuracy is $1.19\%$ across the fifteen conditions (five data-sets with three network architectures per data-set). This difference is small compared to the changes in clean accuracy that can be produced by small changes to the training setup (He et al., 2019; Wightman et al., 2021; Pang et al., 2021). Of the specialized losses, LN loss achieves similar accuracy on clean and corrupt test data as HEM, while DICE and MM losses perform much worse (Fig. 1, $1^{st}$ and $2^{nd}$ segments). Detailed results are given in Appendix C.1.1.

### 4.1.2 PERFORMANCE ON UNKNOWN CLASS REJECTION

Classification accuracy measured on the standard test set, which contains samples from a similar input distribution to the training data, has been the main pre-occupation of most research in the

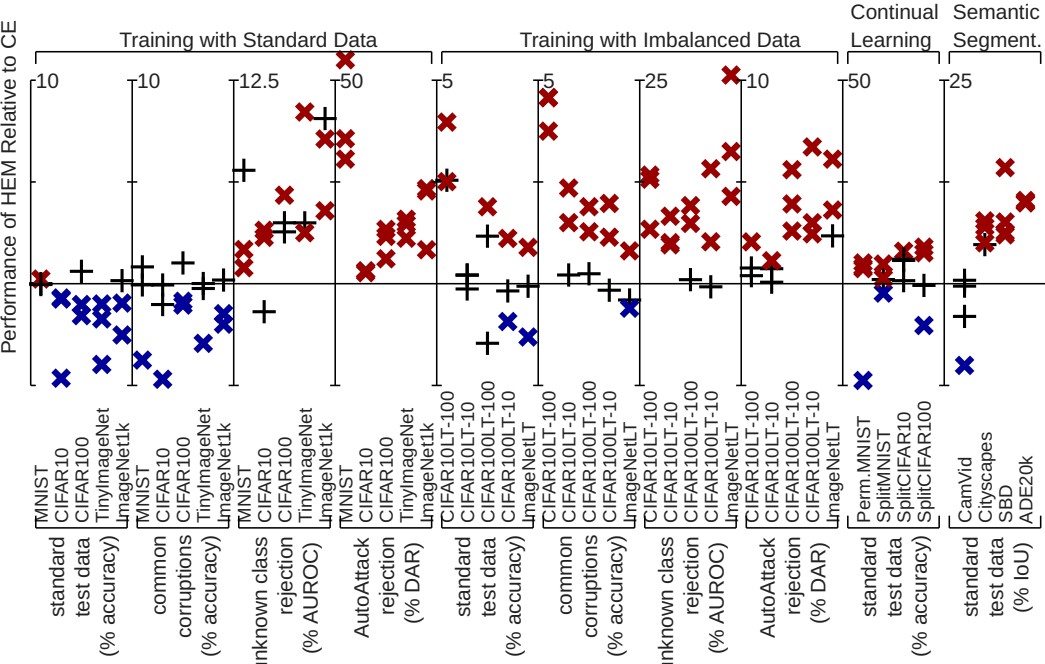

Figure 2: Summary results for all the different tasks considered, showing the relative performance of high error margin (HEM) loss compared to Cross-Entropy (CE) loss. Relative performance is calculated as described in Fig. 1. Hence, points above the horizontal line indicate HEM performance better than that of CE. Please note the separate y-axis scales used in each segment of this figure. In contrast to Fig. 1, here separate results are shown for each training data-set. When there are multiple results for the same data-set, these were obtained using different DNN architectures, or in the case of continual learning different techniques for preventing catastrophic forgetting. Each marker shows the difference in the mean performance achieved across multiple trials. The 'x' and '+' markers indicate experiments where there was or was not a significant statistical difference in the performance produced by the two losses, as evaluated using the two-sample t-test (with p<0.05). The blue/red markers indicate conditions where CE/HEM loss had the significantly better performance. For information about the variability of performance across trials please see the more detailed results in Appendix C.

history of machine learning so far. On such data, as confirmed by the preceding results, CE loss performs the best and for this reason has become the standard loss function for training classifiers. However, more recently, there has been growing concern that accuracy on the standard test data is insufficient to ensure that classifiers are safe, reliable, and trustworthy in more realistic scenarios (Spratling, 2025; Bowers et al., 2023; Amodei et al., 2016; Heaven, 2019; Serre, 2019; Yuille & Liu, 2021; Marcus, 2020; Nguyen et al., 2015; Roy et al., 2022; Sa-Couto & Wichert, 2021; Geirhos et al., 2020; 2018; Ilyas et al., 2019; Papernot et al., 2016; Akhtar & Mian, 2018). In particular, it is well known that CE-trained DNNs are susceptible to making over-confident predictions. For example, when shown samples that do not belong to any of classes in the training data a DNN may predict with high confidence that these samples belong to one of the known categories (Hendrycks & Gimpel, 2017; Amodei et al., 2016; Kumano et al., 2022; Nguyen et al., 2015). Such over-confidence for unknown classes may cause errors in real-world scenarios where such samples might be commonly encountered and in situations where dishonest actors deliberately attempt to fool the classifier into making erroneous predictions.

To evaluate how susceptible a network is to this kind of overconfidence error, we can test whether we can detect and reject out-of-distribution samples based on the confidence of the network (see Appendix B.1.2 for details). This evaluation method is called open-set recognition (Vaze et al., 2022; Yang et al., 2022), out-of-distribution (OOD) detection/rejection (Hendrycks & Gimpel, 2017; Mohseni et al., 2020; Bitterwolf et al., 2022; Zhang & Ranganath, 2023; Hendrycks et al., 2022b),

or unknown class rejection (Spratling, 2025). For the results reported here, Maximum Softmax Probability (MSP; Hendrycks & Gimpel, 2017) was used as the confidence score, but similar results were obtained using Maximum Logit Score (MLS; Vaze et al., 2022; Hendrycks et al., 2022a) (see Fig. 4(c) in Appendix C.1.2).

For unknown class rejection, HEM provides significantly better performance on average than all other tested losses (Fig. 1, $3^{rd}$ segment). In all but one of the fifteen conditions we tested (five data-sets with three network architectures per data-set) HEM outperforms CE (Fig. 2, $3^{rd}$ segment). DICE loss and MM loss perform very badly (Fig. 1, $3^{rd}$ segment). HEM loss even beats LN loss despite LN being a specialised loss that produces state-of-the-art performance on unknown class rejection. Detailed results are given in Appendix C.1.2, and an analysis of the prediction confidence scores produced by different losses is given in Appendix D.2.

### 4.1.3 PERFORMANCE ON AUTOATTACK REJECTION

Another robustness problem faced by DNNs is their susceptibility to adverserial attacks: being fooled into making the wrong prediction by small perturbations that do not change the class of the perturbed sample to a human observer (Szegedy et al., 2014; Goodfellow et al., 2015; Kurakin et al., 2017; Eykholt et al., 2018; Biggio & Roli, 2018). Here we test susceptibility to this problem using the DAR score (Spratling, 2025), which is the proportion of adverserially perturbed samples that are rejected as out-of-distribution or are not rejected but still classified correctly. The rejection/acceptance threshold is set so that 95% of correctly classified clean examples are accepted. Adverserial samples were generated using AutoAttack (AA; Croce & Hein, 2020) (details in Appendix B.1.2).

HEM-trained networks have a large advantage over identical CE-trained architectures in terms of correctly dealing with adversarial attacks (Fig. 1, $4^{th}$ segment). In this case, HEM outperforms CE in all fifteen conditions (Fig. 2, $4^{th}$ segment). Compared to CE, HEM loss has a much larger advantage in terms of its ability to enable accurate unknown class rejection, and also to detect adversarial attacks, than the small disadvantage it has in terms of clean and corrupt accuracy. HEM also outperforms, by a large margin, the other tested losses on adverserial robustness (Fig. 1, $4^{th}$ segment). Detailed results are given in Appendix C.1.2.

### 4.2 LEARNING WITH IMBALANCED DATA-SETS

The ability to learn when the training data contains a very different number of samples for different classes (*i.e.*, with long-tailed data) was tested using the CIFAR10, CIFAR100 and ImageNet training data. This training data was modified to produce long-tailed data, using standard methods used in previous literature, by removing different numbers of samples from each class. Performance was evaluated on multiple network architectures using all the performance criteria used in the Section 4.1 to evaluate networks trained with balanced data-sets. Full details of the experimental methods are provided in Appendix B.2.

A comparison of the performance of networks trained on imbalanced data using CE and HEM losses reveals a similar pattern of results as where obtained with standard training data. Specifically, similar performance for the two losses on standard and corrupt test data (Figs. 1 and 2, $5^{th}$ and $6^{th}$ segments), better performance with HEM loss on unknown class rejection and adversarial attacks (Figs. 1 and 2, $7^{th}$ and $8^{th}$ segments).

Comparing HEM and LA loss (a version of CE designed to improve performance on imbalanced data) shows that LA has an advantage in terms of clean and corrupt accuracy, but that networks trained with HEM are better at identifying, and rejecting, unknown and adversarial samples. The high performance of LA loss on the clean data raises the prospect that there may be more optimal settings for the margins in HEM.

HEM (and HEM-) perform as well as, or better than LN on all of the four evaluation metrics. DICE loss performs significantly worse than HEM on all evaluation criteria. However, the performance of DICE loss on adversarial attacks can, surprisingly, be improved beyond all other losses by using the MLS score as the rejection criterion (see Fig. 6(e) in Appendix C.2). A full set of more detailed results are provided in Appendix C.2.

### 4.3 CONTINUAL LEARNING

The performance of HEM loss when applied to continual learning was assessed using standard benchmark tasks: PermutedMNIST, SplitMNIST, SplitCIFAR10, and SplitCIFAR100. Due to catastrophic forgetting (French, 2003), the over-writing of previously learned weights when training on a new task, all loss functions perform very poorly at continual learning unless a strategy is used to reduce forgetting. Many such strategies have been proposed. Here five were used: Replay (Robins, 1993; Chaudhry et al., 2019), Synaptic Intelligence (SI; Zenke et al., 2017), Elastic Weight Consolidation (EWC; Kirkpatrick et al., 2017), Less-Forgetful Learning (LFL; Jung et al., 2016), and Learning without Forgetting (LwF; Li & Hoiem, 2016). Each loss function was used in combination with each of these continual learning strategies, and performance was evaluated at the end of a sequence of five training episodes using unseen test data for all the five sub-tasks that were learnt during training. Full details of the experimental methods are provided in Appendix B.3

HEM performed considerably better on average at continual learning than CE loss (Fig. 1, $9^{th}$ segment), consistent with our expectations (Section 2.1). In 11 of the 16 conditions tested, better performance was obtained using HEM loss rather than CE loss (Fig. 2, $9^{th}$ segment). Furthermore, if only the best performing combination of loss and strategy of reducing catastrophic forgetting is considered for each loss, then in three of the four tasks HEM loss produces better performance than CE loss (Figs. 7(a) to 7(d) in Appendix C.3). This is remarkable as the training recipes used were designed to produce the best performance for each method of reducing catastrophic forgetting when paired with CE loss.

HEM has even greater advantages over the other applicable losses that were tested: LN, DICE and MM. LN and DICE losses perform very poorly on continual learning. This suggests that, unlike HEM and CE losses, they do not generalise to tasks outside of the specialised domain for which they were developed. As there is no class imbalance in the training data used here, LA is equivalent to CE, and HEM is equivalent to HEM-. Detailed results are given in Appendix C.3.

### 4.4 SEMANTIC SEGMENTATION

Performance on semantic segmentation was assessed using four standard data-sets: CamVid (Brostow et al., 2009), Cityscapes (Cordts et al., 2016), SBD (Hariharan et al., 2011), and ADE20k (Zhou et al., 2017). For each data-set multiple experiments were performed using the FPN architecture (Kirillov et al., 2019) with four different backbones. Full details can be found in Appendix B.4.

HEM- performs better than HEM (Fig. 1, $10^{th}$ segment), suggesting that our heuristic for setting the margins does not generalise from image classification to semantic segmentation. However, even with sub-optimal margins, HEM shows considerably better average performance than CE, and all the other existing losses considered. For some backbone architectures CE loss produced superior image segmentation performance to HEM loss on the CamVid data-set (Fig. 2, $10^{th}$ segment and Fig. 8(a) in Appendix C.4). However, for the three larger data-sets, HEM loss out-performed CE loss with all four backbone architectures (Fig. 2, $10^{th}$ segment and Figs. 8(b) to 8(d) in Appendix C.4).

LN and LA losses perform very poorly (Fig. 1, $10^{th}$ segment) showing that these specialised losses do not work well outside of the specific domain for which they were created. DICE, a specialised loss developed specifically for segmentation tasks, has performance similar to that of CE, but worse than HEM (the proposed loss that uses class specific margins to help deal with imbalanced data) and HEM- (the ablated version of the proposed loss with a single, shared, margin). The advantage of HEM is even clearer if for each data-set only the results for the backbone architecture that gives the best results for each loss is considered (Fig. 8(e) in Appendix C.4).

### 4.5 ABLATION STUDY

HEM, differs from MM loss in terms of 1) using class-specific margins, and 2) how the errors are combined (Section 3). The effects of the first modification can be seen from the results for HEM-that have already been presented. The effects of the second modification were tested using ResNet18 networks trained on CIFAR10 and CIFAR100. The training set-up was as described in Appendix B.1.

The first change to how the errors are combined is to include only above-zero error values when calculating the mean error. This modification alone produces an improvement in classification

Table 2: Ablation study on the effects of the proposed changes to MM loss on classification accuracy. Results are for ResNet18 networks trained on CIFAR10 and CIFAR100 using a margin of $\mu =0.2$. Results are averaged over five trials and the standard deviation is given after the $\pm$ symbol. The best result in each column is highlighted in bold. The changes made to standard MM loss are denoted as "maz" for taking the mean of above-zero errors, and "thres" for setting errors below the mean to zero.

|  | Clean Accuracy (%) | |
| --- | --- | --- |
| Loss | CIFAR10 | CIFAR100 |
| MM | $93.79 \pm 0.11$ | $70.13 \pm 0.19$ |
| +maz | $93.81 \pm 0.23$ | $74.94 \pm 0.35$ |
| +thres | $93.78 \pm 0.22$ | $73.13 \pm 0.26$ |
| +maz+thres = HEM | $\mathbf{93.84 \pm 0.19}$ | $\mathbf{74.95 \pm 0.46}$ |

accuracy (Table 2). As expected, this improvement is greatest for the data-set with the most class labels, as there will be more zero-valued errors across the larger number of logits that this modification enables the loss to ignore.

The second change to how the errors are combined was to set errors less than the mean to zero. On its own this modification is less effective than the first. This is to be expected, as this modification causes even more zeros to be included in the average, causing the loss to become low and learning to cease prematurely. However, when this modification is combined with the first it provides a small additional boost to performance by encouraging the loss to concentrate on the largest errors, particularly at the start of training when there are many errors. The advantage of HEM over MM in terms of clean accuracy is fairly small for the conditions shown in Table 2. However, as shown in Fig. 1, on average, over many data-sets and network architectures, the advantages of HEM over MM are highly significant.

## 5 CONCLUSION

The proposed high error margin (HEM) loss has been shown to performs well across a very wide range of tasks, data-set sizes, and network architectures. It trains classifiers that outperform, or are as good as, those trained with CE loss in all the tasks we have considered except clean and corrupt image classification with balanced training data. Specifically, over the ten different types of evaluation we have performed, corresponding the ten segments of Fig. 2, HEM is superior in eight situations while CE is the best only in two. It is common for there to be a trade-off between clean accuracy and increased robustness (Spratling, 2025), with HEM the sacrifice in clean accuracy is relatively small compared to the large increases in performance on other metrics. Furthermore, the reduction in clean accuracy is likely to be negligible, as optimising the training hyper-parameters can yielded much bigger improvements in clean accuracy than the difference we observe (He et al., 2019; Wightman et al., 2021; Pang et al., 2021). A simple experiment where only the initial learning rate was modified based on intuition gained from observing the learning dynamics substantiates this claim (see Appendix D.3). In all our experiments, our newly proposed loss was at a disadvantage because all training and evaluation choices were optimized for CE loss. This applies to the training hyper-parameters, training schedules, the network architectures, the OOD rejection methods, and the continual learning techniques: all of which have been painstakingly refined over many years to work well with CE loss. Despite these disadvantages HEM almost always performs better than CE loss.

Some specialised losses performed better than CE loss for the tasks they were designed for, but all failed on other tasks. In contrast, HEM loss performed well on all tasks. Comparing all the tested losses across the ten different types of evaluation we have performed, corresponding the ten segments of Fig. 1, HEM is the best performing in five situations while CE is the best only in two. LogitNorm (LN) loss (Wei et al., 2022), performs almost as well as HEM loss at out-of-distribution rejection, but HEM out-performs LN by a considerable margin on continual learning and semantic segmentation. With imbalanced training data, logit-adjusted (LA) loss (Menon et al., 2021) yields better performance on standard test data than HEM loss, but HEM is superior at rejecting out-of-distribution samples. Furthermore, HEM out-performs LA at semantic segmentation, our only other task with class imbalance. For semantic segmentation the commonly used specialised loss, DICE (Milletari et al., 2016), performed no better than CE loss in the experimental set-ups we used. HEM

loss performed semantic segmentation more accurately than DICE, and out-performed DICE by a considerable margin on all other tasks. Following current standard practice we have separately assessed performance against different benchmarks. However, it is not hard to imagine real-world scenarios where multiple advantages of our loss might combine to yield even greater advantages over CE and the specialised losses. For example, a task where it is necessary to learn continuously with long-tailed data and the resulting classifier needs to be robust to unknown classes.

HEM loss is zero for any training sample where the activation of the target logit is sufficiently above the value of the other logits. This means that during the later stages of learning many training samples cause no changes to the network weights, and it is possible for autograd to prune the computational graph. As a result training with HEM is faster than training with CE. For example, it reduces training time by approximately 10% for a ResNet18 trained for 200 epochs on TinyImageNet. In dense prediction tasks there will be fewer opportunities to prune the computational graph, however, we still observe a small reduction in training time when using HEM. For example, training the ResNet34 backbone on Cityscapes was approximately 4% quicker using HEM compared to CE. Future research might explore if, rather than saving time, such zero-loss training samples could be augmented to improve generalisation and/or robustness. More generally, it would be particularly interesting to combine HEM with techniques for improving adversarial robustness, or to see if regularisation terms could be added to HEM loss to improve the representations that are learnt. Additionally, further work might explore alternative heuristics for setting the margins or ways of learning margins for different tasks. Subsequent research might also test HEM in other domains, such as language, as there is no reason why HEM loss should not also work for non-visual classification tasks.

## 6 REPRODUCIBILITY STATEMENT

The experimental set-ups are described in detail in Appendix B. The proposed loss is described fully in Section 3. HEM is trivial to implement and incorporate with an existing code-base. However, to ensure the reproducibility of our results, open-source code implemented in PyTorch (Paszke et al., 2019) which performs all the experiments described in this article will be made publicly available upon publication of this work.

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

# A    EXISTING LOSS FUNCTIONS

For each input image $\mathbf{x}$, a classifier, g, produces a vector of outputs, each element of which is associated with a class label, *i.e.*,: $\mathbf{y} = \mathrm{g}(\mathbf{x})$, where $\mathbf{y} \in \mathbb{R}^n$, and $n$ is the number of classes. For a neural network these outputs are the activations of the neurons in the last layer before applying an activation function. These values are commonly known as the "logits" following a literal interpretation of the cross-entropy loss. The class label, $c$, predicted by such a classifier is that associated with the output with the highest value, *i.e.*, $c = \mathrm{argmax}(\mathbf{y})$.

In addition to predicting the class of the input sample, the classifier can also provide an estimate of its confidence in the classification it has made. Two standard methods of confidence scoring are Maximum Logit Score (MLS; Vaze et al., 2022; Hendrycks et al., 2022a), and Maximum Softmax Probability (MSP; Hendrycks & Gimpel, 2017). MLS is the maximum response of the network before any activation function is applied, *i.e.*, $\max(\mathbf{y})$. MSP is the maximum value of the network output after the application of the softmax activation function, *i.e.*, $\max(\mathbf{z})$ where $\mathbf{z}$ is defined as:

$$z_j = \frac{\exp(\frac{y_j}{\tau})}{\sum_{i=1}^n \exp(\frac{y_i}{\tau})} \tag{3}$$

$\tau$ is a non-negative hyper-parameter that is typically set to a value of one. The softmax function normalises the output values so that they sum to one and can be interpreted as a probability distribution. Smaller values of $\tau$ cause the softmax function to produce a more peaked probability distribution.

The output of the network is also used to define a differentiable loss function that is used to update the parameters so that predictions become more accurate. Those existing loss functions most relevant to this work are described in the following subsections.

## A.1    CROSS-ENTROPY LOSS

Cross-Entropy (CE) loss is defined as:

$$\mathcal{L}_{CE} = -\log z_l \tag{4}$$

where $l$ is the index corresponding to the correct class (*i.e.*, the ground-truth class label), and $\mathbf{z}$ is the output of the softmax activation function applied to the logits (Eq. (3)). As described earlier, the softmax function has a hyper-parameter, $\tau$, which means that CE could be applied using different values of this parameter. However, for almost all applications of CE, $\tau$ is set to a value of one. Hence, a value of $\tau = 1$ was used in all experiments with CE loss described in this paper.

## A.2    LOGITNORM LOSS

LogitNorm (LN) loss is a variation of CE loss that has been shown to produce state-of-the-art results on unknown class rejection when used in conjunction with the Maximum Softmax Probability (MSP) confidence scoring method (Wei et al., 2022). LN loss makes two modifications to CE loss: (1) it normalises the logits by their $l_2$-norm before application of the softmax function, (2) it uses a low value of $\tau$ that causes the softmax function to produce a more peaked probability distribution. Preliminary experiments (Appendix D.1) showed that a hyper-parameter of $\tau = 0.04$ was most effective at unknown class rejection, and hence, that value was used in all experiments with LN loss described in this paper.

### A.2.1    LOGITNORM LOSS AS A MARGIN-LIKE LOSS

The LogitNorm loss yields values close to zero when the output of the correct neuron is sufficiently larger than the other neurons' outputs (Table 1). This behaviour is due to the use of a reduced value of $\tau$. When $\tau$ is sufficiently small the softmax function produces a highly peaked probability distribution and $\frac{\exp(\frac{y_l}{\tau})}{\sum_{i=1}^n \exp(\frac{y_i}{\tau})} \to 1$. This causes LN loss to be zero when the response of the node corresponding to the correct class is sufficiently dominant. Hence, LN loss behaves like a margin loss, as learning stops for samples that are sufficiently well classified. We believe this margin-like behaviour, which prevents increasing confidence, explains the effectiveness of LN loss at distinguishing known from unknown classes.

In contrast, Wei et al. (2022) claim that the effectiveness of LN loss is due to the normalisation of the logits. They believe that normalisation forces learning to generate logit vectors for different classes that are distinct from each other in terms of the angle between them, rather than their magnitude. While we do not believe that normalisation is the primary factor in avoiding over-confidence, normalisation provides other advantages. The normalisation of the logits is responsible for the loss monotonically increasing as the predictions become worse (this is true even when $\tau = 1$). Large negative logits produced by neurons that do not represent the true class (activities that would reduce CE loss) cause a reduction in the normalised logit value associated with the true class, and hence, increase the LN loss. The normalisation of the logits performed by LN also seems to be important to prevent training becoming unstable: we found that LN loss was capable of successfully training networks with small $\tau$ values, while the same small $\tau$ values would cause CE loss to fail. The cause of this instability is possibly that a low value of $\tau$ can result in the loss being very large when the prediction is very wrong (see last row of Table 1), a situation that is common early in training when the network's outputs are random.

### A.3 LOGIT-ADJUSTED LOSS

Logit-adjusted (LA) loss is a variation of CE loss that is designed to produce improved performance when training with imbalanced data (Menon et al., 2021). Before the application of the softmax function, the logits are modified by a term that is proportional to the relative number of training samples in each class. Hence,

$$\mathcal{L}_{LA} = -\log z'_l \tag{5}$$

where:

$$z'_j = \frac{\exp(y_j + \log(p_j))}{\sum_{i=1}^n \exp(y_i + \log(p_i))} \tag{6}$$

The term $p_j$ is the proportion of training samples in class $j$, i.e. $p_j = s_j / \sum_{i=1}^n s_i$ where $s_j$ is the number of samples in class $j$. A number of similar losses have been proposed which use alternative methods to adjust the logits (Tan et al., 2020; Ren et al., 2020; Cao et al., 2019). However, LA loss has been found to produce better results than these alternatives and other methods of dealing with class imbalance (Menon et al., 2021; Zhao et al., 2024). Note, that for balanced data-sets, $p_j$ has the same value for each class and LA loss is identical to CE loss.

### A.4 DICE LOSS

DICE loss (Milletari et al., 2016) is an alternative to CE that is frequently used for image segmentation tasks (Azad et al., 2023; Ma et al., 2021). It uses a measure of the overlap between the one-hot encoded target outputs, $\mathbf{t}$, and the softmax predictions, $\mathbf{z}$, such that:

$$\mathcal{L}_{DICE} = 1 - 2\frac{\sum_i(t_i \times z_i)}{\sum_i(t_i + z_i)} \tag{7}$$

In multi-class applications, DICE loss is calculated separately for each class (the sums in Eq. (7) are taken over the samples in the batch), and the overall DICE loss is the mean of the separate class losses.

### A.5 MULTI-CLASS MARGIN LOSS

Multi-class Margin (MM) loss, also known as the classification hinge loss (Crammer & Singer, 2002), defines an error, $e_i$, associated with each logit, $y_i$, as follows:

$$e_i = \begin{cases} \max(0, y_i - y_l + \mu) & \text{if } i \neq l \\ 0 & \text{if } i = l \end{cases} \tag{8}$$

where $\mu$ is the margin, a non-negative hyper-parameter. MM loss combines the error for different logits (and across all samples in a batch) by taking the mean, so that:

$$\mathcal{L}_{MM} = \frac{1}{n}\sum_{i=1}^n e_i \tag{9}$$

Preliminary experiments (Appendix D.1) showed that the value of the margin had little influence on classification accuracy, but had a stronger influence on unknown class rejection performance. A value of $\mu = 1$ was used in all subsequent experiments as this was most effective at unknown class rejection and is the default value typically used for this loss.

# B EXPERIMENTAL METHODS

## B.1 LEARNING WITH STANDARD DATA-SETS

### B.1.1 TRAINING DATA

Performance was assessed for DNNs trained on standard image classification data-sets: MNIST (LeCun et al., 1998), CIFAR10 (Krizhevsky, 2009), CIFAR100 (Krizhevsky, 2009), TinyImageNet (TIN) and ImageNet1k (IN Russakovsky et al., 2015)(IN). These data-set vary in terms of the size of the images (from 28-by-28 pixels with 1 colour channel to 224-by-224 pixels with 3 colour channels), the number of categories (10 to 1000), and the number of training samples (from 50k to 1.28M). For all datasets, standard data augmentations were applied to the training images: horizontal flipping and random cropping for both CIFAR data-sets and TinyImageNet, horizontal flipping, resizing to 256 pixels, and a centre crop for ImageNet1k. For all data-sets the standard split between training and testing exemplars was employed. Pixel values in both the training and testing samples were scaled to the range [0,1].

### B.1.2 PERFORMANCE METRICS

Performance was evaluated against a number of different criteria.

**Performance on standard test data**  Firstly, the percentage of samples correctly classified from the standard test set provided with each training data-set was calculated (the "clean" accuracy).

**Performance on common corruptions data**  Secondly, the ability of trained networks to generalise to input distribution shifts was assessed by determining classification accuracy with the common corruptions data-sets: MNIST-C, CIFAR10-C, CIFAR100-C, TinyImageNet-C and ImageNet-C (Hendrycks & Dietterich, 2019; Mu & Gilmer, 2019). MNIST-C contains 15 different corruptions including different types of noise, blurring, geometric transformations, and superimposed patterns. The others contain 18 different corruptions including different types of noise, blurring, synthetic weather conditions, and digital corruptions. As is typical in the literature, performance was evaluated by averaging performance over all the corruptions at all degrees of intensity.

**Performance on unknown class rejection**  A third performance metric was used to assess the ability of a network to distinguish known from unknown classes. This was evaluated using the Area Under the Receiver Operating Characteristic curve (AUROC) as this is a common choice in the literature (Kirchheim et al., 2022; Chen et al., 2023; Cheng et al., 2023; Xu-Darme et al., 2023; Yang et al., 2022; 2023; Lee et al., 2022). AUROC is calculated separately for each unknown class data-set, evaluating how distinct the confidence scores produced by samples from the standard test-set are from the confidence scores produced in response to samples from the unknown class data-set. The standard, baseline, method for determining confidence uses the maximum value of the network output after the application of the softmax activation function (*i.e.*, $\max(\mathbf{z})$). As a result it is called Maximum Softmax Probability (MSP; Hendrycks & Gimpel, 2017). MSP was used by default in our assessment, but some evaluations were repeated using an alternative: Maximum Logit Score (MLS; Vaze et al., 2022; Hendrycks et al., 2022a). MLS defines the confidence that a sample is of a known class as the maximum response of the network output before any activation function is applied (*i.e.*, $\max(\mathbf{y})$).

AUROC was calculated using seven data-sets containing unknown classes, and the average AUROC across all seven data-sets was reported. The seven data-set used to evaluate networks trained with MNIST were the test-sets of Omniglot (Lake et al., 2015), FashionMNIST (Xiao et al., 2017), KM-NIST (Clanuwat et al., 2018) and four data-sets containing synthetic images: (1) images containing random blobs, as used in (Hendrycks et al., 2019); (2) images in which each pixel intensity value was independently and randomly selected from a uniform distribution; (3) the images of the standard

(clean) test set after a random permutation of all pixels; (4) the images of the clean test set after randomising the phase, in the Fourier domain, of each image. Each of these four synthetic data-sets contained 10000 samples. The CIFAR10 trained networks were tested using unknown classes from the test-sets of the Textures (Cimpoi et al., 2014), SVHN (Netzer et al., 2011), and CIFAR100 data-sets, plus, four synthetic image data-sets generated as described before. For CIFAR100 trained networks the same seven OOD data-sets were used as for CIFAR10, except CIFAR10 was used in place of CIFAR100. Networks trained on TinyImageNet and ImageNet1k were evaluated using Textures (Cimpoi et al., 2014), the iNaturalist 2021 validation set (Van Horn et al., 2018), the ImageNet-O data-set (Hendrycks et al., 2021), and the four synthetic image data-sets generated as described previously.

**Performance on AutoAttack rejection**     Finally, performance was also evaluated using adversarial attacks generated using AutoAttack (AA; Croce & Hein, 2020), a state-of-the-art ensemble attack method that employs both gradient-based (white-box) and gradient-free (black-box) attacks. AA was implemented using the torchattacks PyTorch library (Kim, 2021). Two sets of adversarial samples where created. Each set was created by perturbing 10000 samples from the standard (clean) test-set, but with a different method of constraining the magnitude of the perturbation. Specifically, AA was used to apply both $l_\infty$ and $l_2$-norm constrained attacks. The perturbation budget ($\epsilon$) used for each attack was the standard value used in the previous literature for each data-set. Specifically, $\epsilon$ was set to $\frac{8}{255}$ and 0.5 for $l_\infty$ and $l_2$-norm attacks, respectively, against networks trained on CIFAR10, CIFAR100, TinyImageNet, and ImageNet1k, and $\epsilon$ was set to 0.3 for $l_\infty$-norm and to 2 for $l_2$-norm attacks on MNIST trained networks.

Networks were not trained to be able to correctly classify adversarial examples, and hence, robust accuracy was low for all the evaluated losses. However, networks can still be robust if they are capable of identifying, and rejecting samples that have been adversarially perturbed. Adversarial robustness was evaluated using detection accuracy rate (DAR; Spratling, 2025). DAR determines the proportion of samples that are processed correctly. Where for adversarial samples, "processed correctly" means that the sample is accepted and the predicted class label is correct, or it is rejected and the predicted class label is wrong (Zhu et al., 2024). As for unknown class rejection, a sample is accepted or rejected based on the confidence of the prediction made by the network under evaluation. Confidence was measured using either Maximum Softmax Probability (MSP) or Maximum Logit Score (MLS), and the threshold used to reject samples was set such that 95% of correctly classified samples from the standard test set were accepted (Zhu et al., 2024).

### B.1.3   NEURAL NETWORK ARCHITECTURES

A large variety of DNNs architectures were used as summarised in Table 3. A small version of LeNet (LeCun et al., 1998) with 16 channels in the two convolutional layers, and 50 neurons in the penultimate, fully-connected, layer. A simple, fully-convolutional neural network (ConvNet) consisting of 5 convolutional layers, each containing 32 3-by-3 masks and using the ReLU activation function. This architecture performed down-sampling using average pooling and it did not use batch (or any other form of) normalisation. It is a simple, sequential, hierarchy without any parallel pathways or skip connections. A simple fully-connected network (MLP) consisting of three hidden layers each containing 200 neurons and employing the ReLU activation function. ResNets (He et al., 2016a), specifically, ResNet18, ResNet32, and ResNet50. WideResNets (Zagoruyko & Komodakis, 2016), specifically, WRN22-10 and WRN28-10. PreActResNet18 (PARN18; He et al., 2016b). MobileNet version 3 (Howard et al., 2019; 2017), specifically the small model (MobileNetS) and the large model (MobileNetL). The inception architecture version 3 (Szegedy et al., 2016; 2015). The Swin Transformer (version 2) (Liu et al., 2021; 2022a) tiny (SwinT). The vision transformer (Dosovitskiy et al., 2020) base model with 16×16 input patch size (ViT-B/16).

Our implementations of ResNets, WRNs, and PARN were based on the code provided with (Pang et al., 2021),[1] except for the implementation of ResNet32 which was adapted from code by Yerlan Idelbayev,[2] and ResNet50 which came from the PyTorch Hub.[3] The implementations of MobileNetv3, inception3, and the Transformers were also from the PyTorch Hub. The inception3 was modified to

---

[1] https://github.com/P2333/Bag-of-Tricks-for-AT
[2] https://github.com/akamaster/pytorch_resnet_cifar10/
[3] https://pytorch.org/vision/stable/models.html

Table 3: A summary of the neural network architectures used to assess performance on image classification when learning with standard data-sets. For each model the number of trainable parameters is indicated in brackets. For each data-set the architectures are arranged from left-to-right in order of increasing size. Note that ResNet50 and ResNet18 are large networks designed for use with ImageNet1k (but using a different stem when applied to smaller images), while ResNet32 is a smaller network designed for use with CIFAR10, and hence, has fewer parameters than ResNet18 despite its greater depth.

| Data-set | Model 1 | Model 2 | Model 3 |
|---|---|---|---|
| MNIST | LeNet (20,194) | ConvNet (30,954) | MLP (239,410) |
| CIFAR10 | ResNet32 (464,154) | MobileNetS (1,528,106) | WRN22-10 (27,977,146) |
| CIFAR100 | MobileNetL (4,330,132) | ResNet18 (11,220,132) | PARN18 (11,218,340) |
| TinyImageNet | ResNet18 (11,173,962) | inception (23,995,504) | WRN28-10 (38,241,656) |
| ImageNet1k | ResNet50 (25,557,032) | SwinT (28,351,570) | ViT-B/16 (86,567,656) |

allow it to work with TinyImageNet as follows. Before the inception layers the original architecture, designed for use with the larger images in ImageNet1k, contains three standard convolution layers, a max pooling layer, two further standard convolution layers, and another max pooling layer. Both maxpooling and the two convolution layers between them were removed. Furthermore, the size of the filters in the second convolutional layer in the auxiliary head were changed from 5-by-5 to 4-by-4.

### B.1.4 TRAINING SETTINGS

To ensure a fair comparison between loss functions, while avoiding the need to search for optimal training hyper-parameters for each combination of loss function, network architecture and data-set, the same training hyper-parameters were used for all the experiments performed using the same combination of data-set and network architecture. In general, five repeats were made of each experiment, except those experiments performed with ImageNet1k where either three repeats (when using the two smaller models listed in Table 3) or one experiment (using the largest model listed in Table 3) were performed.

**MNIST**  For all experiments with MNIST, training was performed for 20 epochs using the Adam optimiser (Kingma & Ba, 2015) with a batch size of 128 and a fixed learning rate of $10^{-3}$. This set-up was found in preliminary experiments to be adequate for obtaining high test-set accuracy when training ConvNet with CE loss.

**CIFAR10 and CIFAR100**  For training with both CIFAR data-sets, the SGD optimiser was used for 110 epochs with a momentum of 0.9, a batch size of 128, weight decay of 5e-4, an initial learning rate of 0.1 and a step-wise learning schedule reducing the learning rate by a factor of 10 at epochs 100 and 105. This set-up was taken from (Pang et al., 2021) where it was found to be optimal for the adversarial-training of networks using CE loss. As we are not using adversarial training, this set-up is probably sub-optimal for all the loss functions we compare. If it does favour one loss function, that is likely to be CE. Using this training setup with MobileNets resulted in poor results: CIFAR100 clean accuracy of less than 50% with all loss functions, and chance accuracy on one trial with LN loss. A search for a better initial learning rate (with all other learning hyper-parameters as described before), performed using CE loss and CIFAR100, found that a value of 0.02 produced the best performance. This lower initial learning rate was therefore used for all experiments with MobileNet.

**TinyImageNet**  For training networks on TinyImageNet more epochs are required to reach reasonable performance. Hence, compared to the settings for CIFAR, the number of epochs was increased to 200. Furthermore, the training schedule was changed to decay (by a factor of 10) the learning rate at 50, 100, and 150 epochs. Except for an additional learning rate decay at 50 epochs, the resulting set-up is identical to that used in (Rice et al., 2020), for adversarially-training networks with CE loss.

**ImageNet1k**  With the ImageNet1k data-set the ResNet50 architecture was trained using SGD for 100 epochs with a momentum of 0.9, a batch size of 512, weight decay of 1e-4, and an initial learning rate of 0.1 that decayed by a factor of 10 at epochs 25, 50, and 75. This recipe uses a larger batch

size, 10 more epochs, and one more learning rate decay, but is otherwise the same as baseline training method typically used for training ResNet50 on ImageNet1k.[4] For training the Swin Transformer on ImageNet1k the set-up was based on that proposed in Irandoust et al. (2022). Namely, using the AdamW optimiser with a fixed learning rate of $10^{-3}$ preceded by an exponential learning-rate warm-up period of five epochs.[5] Training was performed for 100 epochs with a batch size of 256. This same set-up, but with a 10 epoch warm-up period, was used to train the ViT-B/16 architecture.

## B.2 LEARNING WITH IMBALANCED DATA-SETS

### B.2.1 TRAINING DATA

Learning with imbalanced training data was assessed using long-tailed versions of CIFAR10, CI-FAR100 and ImageNet (Cao et al., 2019; Liu et al., 2019; Wang et al., 2021). These are standard benchmark tasks in this domain, where the training data is generated from the original, balanced, data-sets by removing samples unequally from each class. Specifically, each long-tailed data-set is created by taking only the first $s_j \times f^j$ samples for the class with index $j$ ($j \in \{0, \ldots, n-1\}$). $f$ is a factor that determines the degree of imbalance. From CIFAR10 two long-tailed data-sets were created using $f = 0.6$ and $f = 0.7744$, and from CIFAR10 two long-tailed data-sets were created using $f = 0.955$ and $f = 0.9771$. The first value of $f$ for each data-set generates a long-tailed set in which the ratio of the number of samples in the classes with the largest and smallest numbers is 100. The second values of $f$ produce an imbalance ratio of 10. For ImageNetLT the standard image sub-sets were used[6] which define an imbalance ratio of 256.

### B.2.2 PERFORMANCE METRICS

Performance was assessed using standard, balanced, test-data sets. The same range of evaluation metrics were used as were used to assess the performance of networks trained on standard training data-sets (as described in Appendix B.1.2).

### B.2.3 NEURAL NETWORK ARCHITECTURES

Each of the four long-tailed CIFAR data-sets were used to train three architectures: ResNet32, ResNet18, and WideResNet20-10. ImageNetLT was used to train ResNet18, SwinT, and ConvNeXt-tiny (Liu et al., 2022b).

### B.2.4 TRAINING SETTINGS

Five repeats were performed of each experiment (combination of loss, network architecture, and training data-set). For the CIFAR data-sets, the training set-up was based on that used in previous work with the same training data (Cao et al., 2019; Cui et al., 2019). Specifically, networks were trained for 200 epochs using SGD with a momentum of 0.9, a batch size of 128, weight decay of 2e-4, and an initial learning rate of 0.1, that was reduced by a factor of 100 at the end of epochs 160 and 180. For ImageNetLT the training set-up was that same as that used for ImageNet as described in Appendix B.1.4.

## B.3 CONTINUAL LEARNING

### B.3.1 TRAINING DATA

Performance on continual learning was assessed with the aid of the Avalanche library (Lomonaco et al., 2021; Carta et al., 2023) using four standard benchmark tasks: PermutedMNIST, SplitMNIST, SplitCIFAR10, and SplitCIFAR100. In each case models were trained on a sequence of five sub-sets of data (training "episodes"). Each trial used a different, randomly selected, sequence of training data

---

[4]https://pytorch.org/blog/how-to-train-state-of-the-art-models-using-torchvision-latest-primitives/

[5]The warm-up period was extended to 10 epochs when using CE loss, as training with the original set-up resulted in a training collapse and final training set accuracy at chance level.

[6]https://drive.google.com/drive/u/1/folders/1j7Nkfe6ZhzKFXePHdsseeeGI877Xu1yf

sub-sets, and this same sequence of sub-tasks was used with each loss to ensure a fair comparison. Each loss function was tested in combination with five strategies for reducing catastrophic forgetting: Replay (Robins, 1993; Chaudhry et al., 2019), Synaptic Intelligence (SI; Zenke et al., 2017), Elastic Weight Consolidation (EWC; Kirkpatrick et al., 2017), Less-Forgetful Learning (LFL; Jung et al., 2016), and Learning without Forgetting (LwF; Li & Hoiem, 2016). The tasks and continual learning strategies were chosen as code for implementing them was available in the Continual Learning Baselines repository.[7]

### B.3.2 PERFORMANCE METRICS

Performance was evaluated at the end of training by measuring classification accuracy with an unseen test set containing equal numbers of samples from each sub-task.

### B.3.3 NEURAL NETWORK ARCHITECTURES

For each combination of task and continual learning strategy we used the same neural network architecture as used in the Continual Learning Baselines repository. For the MNIST tasks the networks were MLPs, while for the CIFAR tasks the architecture was a ResNet18.

### B.3.4 TRAINING SETTINGS

Five repeats were performed of each experiment (combination of loss, continual learning strategy, and task). For each combination of task and continual learning strategy we used the same training set-up as used in the Continual Learning Baselines repository. Where this repository only provided a training recipe for MNIST (or CIFAR10/100), we altered it for use with the other data-sets only by changing the number of epochs (so that the number was 10 times larger for the CIFAR data-sets than for MNIST).

## B.4 SEMANTIC SEGMENTATION

### B.4.1 TRAINING DATA

Performance on semantic segmentation was assessed with the aid of the Pytorch Segmentation Models Library[8] using four data-sets: CamVid (Brostow et al., 2009), Cityscapes (Cordts et al., 2016), and SBD (Hariharan et al., 2011), and ADE20k (Zhou et al., 2017).

### B.4.2 PERFORMANCE METRICS

In each case, performance was evaluated using the mean percentage intersection-over-union (IoU) metric.

### B.4.3 NEURAL NETWORK ARCHITECTURES

All experiments were performed with the FPN architecture (Kirillov et al., 2019) using four different networks as the encoder-backbone: ResNet34 (He et al., 2016a), EfficientNet-b4 (Tan & Le, 2019), DenseNet201 (Huang et al., 2017), and ResNeXt50 (Xie et al., 2017).

### B.4.4 TRAINING SETTINGS

Five repeats were made of each experiment (combination of loss, backbone, and training data-set), except those experiments performed with ADE20k where four repeats were performed.

The training set-up was based on that used previously for training on the CamVid data-set (Badrinarayanan et al., 2017). Specifically, SGD with momentum of 0.9 was used with a fixed learning rate of 0.1 and a batch size of 12. As not all data-sets contain separate test and validation data a fixed number of training epochs was used, rather than selecting the best checkpoint as was done by Badrinarayanan et al. (2017). 100 epochs was used for CamVid, 50 epochs were used for Cityscapes

---

[7]https://github.com/ContinualAI/continual-learning-baselines
[8]https://github.com/qubvel/segmentation_models.pytorch

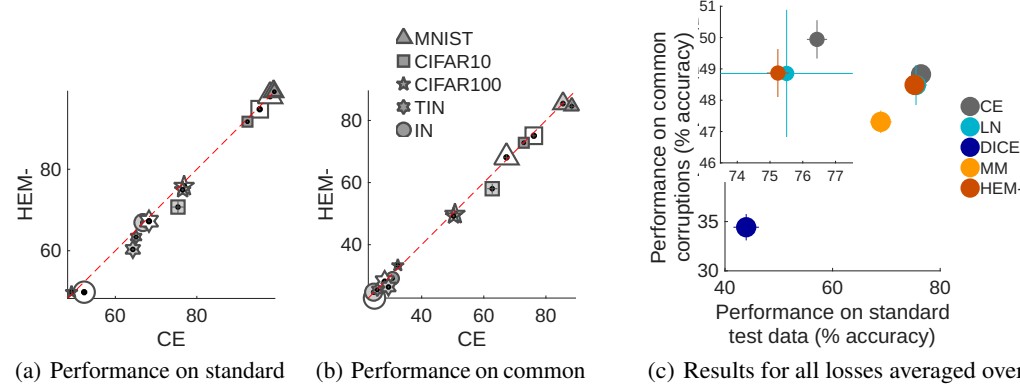

(a) Performance on standard test data (% accuracy).

(b) Performance on common corruptions (% accuracy).

(c) Results for all losses averaged over conditions.

Figure 3: Results when learning with standard data-sets and testing with clean and corrupt images. (a) and (b) directly compare the performance produced by HEM and cross-entropy (CE) losses when used to train networks with MNIST, CIFAR10, CIFAR100, TinyImageNet (TIN), and ImageNet1k (IN) using three different network architectures for each data-set. For each data-set the size of the marker used corresponds to the size of the network. Results above the diagonal are conditions where better performance was obtained when training with HEM rather than CE loss. Performance metrics are averaged over multiple trials performed for each condition (data-set and architecture) and the error bars show the standard deviation recorded across the trials in each condition (in the majority of cases these error bars are too small to be visible). (a) Compares the performance of the two losses in terms of the accuracy of classifying the standard test-data. (b) Compares the performance of the two losses in terms of the accuracy of classifying the common-corruptions test-data. (c) Shows results averaged over all the data-sets and network architecture (and multiple trials in each condition) for all relevant losses: cross-entropy (CE), LogitNorm (LN), DICE, multi-class margin (MM) and HEM. Error bars show the mean standard deviation recorded across the trials in each condition. The inset shows the results for CE, LN, and HEM losses plotted on a separate scale to allow the differences between these losses to be visible.

and SBD, and 20 epochs for ADE20k. For the larger data-sets (Cityscapes, SBD, and ADE20k) the batch size was reduced to four in order to fit within GPU memory.

For the SBD data-set, CE loss failed to learn when using a learning rate of 0.1. A hyper-parameter search was therefore carried out using CE loss to test alternative learning rates (0.05, 0.02, 0.005, 0.001). A learning rate of 0.05 was found to work best with CE loss, so this learning rate was used in all experiments with all losses and the SBD data-set. A learning rate of 0.05 was also used for all experiments with the ADE20k where it was found that the performance of CE loss was unaffected across a range (0.1, 0.05, 0.02, 0.01) of different learning.

# C    DETAILED EXPERIMENTAL RESULTS

## C.1    LEARNING WITH STANDARD DATA-SETS

### C.1.1    PERFORMANCE ON STANDARD TEST DATA AND COMMON CORRUPTIONS DATA

Detailed results showing the absolute, rather than relative, performance of CE and HEM trained networks for each individual condition together with error-bars can be seen in Fig. 3(a) for the standard test data, and in Fig. 3(b) the common corruptions data. This is the data summarised in the $1^{st}$ and $2^{nd}$ segments of Fig. 2. A comparison of the performance of all tested losses is provided in Fig. 3(c). The same results appear in the $1^{st}$ and $2^{nd}$ segments of Fig. 1. The numerical data can be found in the columns headed "Clean" and "Corrupt" in Table 4.

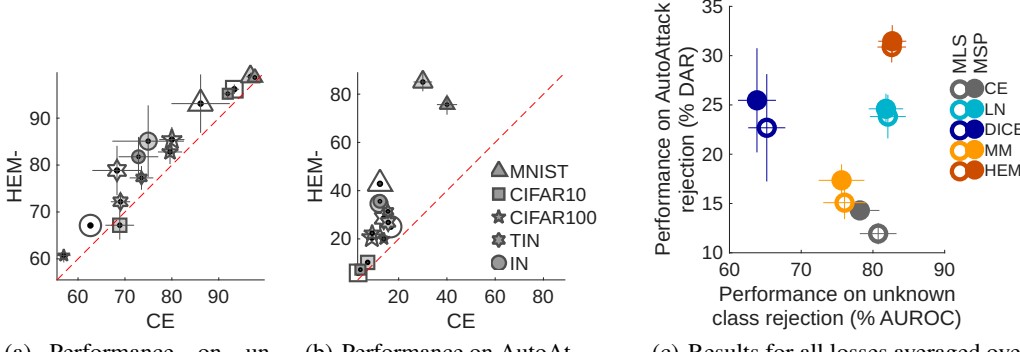

(a) Performance on unknown class rejection (% AUROC).

(b) Performance on AutoAttack rejection (% DAR).

(c) Results for all losses averaged over conditions.

Figure 4: Results when learning with standard data-sets and testing on unknown and adversarial images. This figure has an identical format to Fig. 3 except (a) compares the performance of CE and HEM losses in terms of the ability to distinguish samples from known and unknown classes, and (b) compares the performance of CE and HEM losses in terms of the ability to deal correctly with adversarially perturbed samples. In both (a) and (b) Maximum Softmax Probability (MSP) is used as the confidence score. (c) Shows results averaged over all the data-sets and network architecture (and multiple trials in each condition) for all relevant losses. Closed markers indicate that Maximum Softmax Probability (MSP) was used as the confidence score, while open markers plot results when using Maximum Logit Score (MLS).

### C.1.2 PERFORMANCE ON UNKNOWN CLASS REJECTION AND AUTOATTACK REJECTION

Detailed results showing the absolute, rather than relative, performance of CE and HEM trained networks for each individual condition together with error-bars can be seen in Fig. 4(a) for unknown class rejection, and in Fig. 4(b) for AutoAttack rejection. This is the data summarised in the $3^{rd}$ and $4^{th}$ segments of Fig. 2. A comparison of the performance of all tested losses is provided in Fig. 4(c). The same results appear in the $3^{rd}$ and $4^{th}$ segments of Fig. 1. The numerical data can be found in the columns headed "OOD" and "AA" in Table 4.

## C.2 LEARNING WITH IMBALANCED DATA-SETS

### C.2.1 PERFORMANCE ON STANDARD TEST DATA AND COMMON CORRUPTIONS DATA

A detailed comparison of the performance of CE and HEM- (the ablated version of HEM that uses a single margin, and hence, like CE has no additional mechanism for dealing with class imbalance) is provided in Figs. 5(a) and 5(b) for the standard test data, and the common corruptions data, respectively. The corresponding detailed comparisons of HEM and LA losses is provided in Figs. 5(c) and 5(d). The $5^{th}$ and $6^{th}$ segments Fig. 2 show the performance of HEM relative to CE for the standard test data, and the common corruptions data, respectively.

A comparison of results for all tested losses, averaged over the four conditions (and 5 trials per condition) can be seen in Fig. 5(e). The same results appear in the $5^{th}$ and $6^{th}$ segments of Fig. 1. The numerical data can be found in the columns headed "Clean" and "Corrupt" in Table 5.

### C.2.2 PERFORMANCE ON UNKNOWN CLASS REJECTION AND AUTOATTACK REJECTION

A detailed comparison of the performance of CE and HEM- is provided in Figs. 6(a) and 6(b) for unknown class rejection and adversarial sample rejection, respectively. The corresponding detailed comparisons of HEM and LA losses is provided in Figs. 6(c) and 6(d). The $7^{th}$ and $8^{th}$ segments Fig. 2 show the performance of HEM relative to CE for unknown class rejection and adversarial sample rejection, respectively.

Table 4: A comparison of the performance produced by different losses when applied to learning with standard data-sets. Bold text indicates the best performance on each metric for each combination of training data-set and network architecture.

| Task | Loss | Clean Acc. (%) | Corrupt Acc. (%) | OOD AUROC (%) | AA DAR (%) | Clean Acc. (%) | Corrupt Acc. (%) | OOD AUROC (%) | AA DAR (%) | Clean Acc. (%) | Corrupt Acc. (%) | OOD AUROC (%) | AA DAR (%) |
|---|---|---|---|---|---|---|---|---|---|---|---|---|---|
| **MNIST** | | *LeNet* | | | | *ConvNet* | | | | *MLP* | | | |
| | CE | 99.02 ± 0.05 | **88.33 ± 1.20** | 97.69 ± 0.26 | 40.04 ± 4.19 | 98.84 ± 0.07 | **85.50 ± 0.99** | 96.74 ± 0.84 | 30.02 ± 3.85 | **97.97 ± 0.12** | 67.29 ± 0.91 | 86.13 ± 6.22 | 12.24 ± 0.71 |
| | LN | 98.88 ± 0.21 | 81.85 ± 1.66 | 97.56 ± 0.80 | 72.33 ± 9.93 | 99.02 ± 0.11 | 83.18 ± 1.87 | 97.18 ± 1.45 | 82.49 ± 4.36 | 97.95 ± 0.07 | **68.35 ± 0.90** | 89.48 ± 1.88 | 32.57 ± 1.96 |
| | DICE | 98.83 ± 0.10 | 88.15 ± 0.72 | 95.70 ± 0.53 | 53.61 ± 3.77 | 98.94 ± 0.31 | 85.16 ± 2.15 | 88.58 ± 0.70 | 37.60 ± 6.76 | 97.16 ± 0.58 | 67.30 ± 1.11 | 79.68 ± 2.14 | 24.80 ± 2.89 |
| | MM | **99.07 ± 0.08** | 87.93 ± 1.61 | 97.91 ± 0.34 | 41.14 ± 5.73 | 98.94 ± 0.10 | 84.34 ± 1.65 | 97.08 ± 0.82 | 30.45 ± 2.68 | 97.80 ± 0.23 | 67.75 ± 0.58 | 90.20 ± 4.66 | 12.33 ± 1.11 |
| | HEM- | 99.03 ± 0.14 | 84.57 ± 2.09 | **98.67 ± 0.38** | **75.66 ± 6.14** | **99.07 ± 0.10** | 85.44 ± 1.71 | **98.84 ± 0.37** | **85.04 ± 3.65** | 97.93 ± 0.21 | 68.12 ± 1.44 | **93.10 ± 0.64** | **42.85 ± 5.23** |
| **CIFAR10** | | *ResNet32* | | | | *MobileNetS* | | | | *WRN22-10* | | | |
| | CE | **92.43 ± 0.12** | 72.88 ± 0.59 | 91.92 ± 1.16 | 4.14 ± 0.28 | **75.37 ± 1.42** | **62.71 ± 1.62** | **68.89 ± 3.06** | 7.15 ± 0.30 | **95.43 ± 0.18** | 76.10 ± 0.67 | 93.31 ± 1.01 | 3.21 ± 0.20 |
| | LN | 92.23 ± 0.12 | **73.04 ± 0.59** | **95.40 ± 0.45** | 3.89 ± 0.29 | 58.36 ± 27.31 | 48.46 ± 21.89 | 62.19 ± 7.11 | 10.11 ± 0.39 | 95.11 ± 0.12 | **76.43 ± 0.53** | **97.09 ± 0.21** | 4.39 ± 0.52 |
| | DICE | 86.64 ± 0.23 | 67.16 ± 0.42 | 88.07 ± 0.95 | 5.50 ± 0.20 | 70.12 ± 0.67 | 59.02 ± 0.60 | 63.38 ± 1.97 | **12.34 ± 1.10** | 92.42 ± 0.27 | 73.61 ± 0.23 | 90.04 ± 1.43 | 4.36 ± 0.28 |
| | MM | 91.54 ± 0.14 | 70.66 ± 0.60 | 92.49 ± 1.53 | 3.96 ± 0.25 | 72.90 ± 0.98 | 61.03 ± 1.35 | 67.82 ± 2.49 | 8.78 ± 0.46 | 94.85 ± 0.12 | 74.31 ± 0.78 | 93.51 ± 0.89 | 4.07 ± 0.36 |
| | HEM- | 91.67 ± 0.29 | 72.82 ± 0.55 | 95.20 ± 0.71 | **7.26 ± 0.71** | 70.72 ± 0.71 | 58.02 ± 0.86 | 67.17 ± 2.10 | 10.26 ± 0.42 | 94.73 ± 0.06 | 75.08 ± 0.86 | 96.16 ± 0.48 | **5.91 ± 0.38** |
| **CIFAR100** | | *MobileNetL* | | | | *ResNet18* | | | | *PARN18* | | | |
| | CE | 49.25 ± 0.96 | 32.31 ± 1.02 | 56.96 ± 1.44 | 13.86 ± 0.70 | **76.52 ± 0.29** | **50.27 ± 0.40** | 79.61 ± 2.57 | 9.01 ± 0.36 | **76.87 ± 0.23** | **50.71 ± 0.29** | 79.99 ± 2.68 | 8.90 ± 0.37 |
| | LN | 49.73 ± 0.71 | 33.11 ± 0.53 | 59.41 ± 2.44 | 14.04 ± 0.78 | 76.19 ± 0.28 | 49.90 ± 0.22 | 80.51 ± 3.14 | 15.75 ± 0.98 | 76.15 ± 0.21 | 50.70 ± 0.28 | 84.90 ± 0.84 | 15.93 ± 0.47 |
| | DICE | 2.20 ± 0.40 | 2.03 ± 0.35 | 44.79 ± 7.42 | **42.48 ± 18.16** | 66.96 ± 0.11 | 44.81 ± 0.31 | 73.97 ± 2.46 | 12.10 ± 0.64 | 18.75 ± 1.50 | 15.35 ± 1.12 | 55.93 ± 1.52 | **61.32 ± 5.36** |
| | MM | 40.34 ± 0.37 | 26.17 ± 0.43 | 56.96 ± 4.11 | 13.97 ± 0.78 | 70.59 ± 0.30 | 45.20 ± 0.32 | 80.24 ± 4.27 | 10.85 ± 0.71 | 70.28 ± 0.22 | 45.95 ± 0.27 | 79.29 ± 5.88 | 11.73 ± 0.19 |
| | HEM- | **49.86 ± 0.90** | **33.33 ± 0.80** | **60.70 ± 4.24** | 19.99 ± 0.82 | 74.95 ± 0.46 | 49.22 ± 0.26 | **82.79 ± 2.33** | **22.36 ± 1.27** | 75.85 ± 0.16 | 49.84 ± 0.37 | **85.43 ± 2.20** | 20.52 ± 1.13 |
| **TIN** | | *ResNet18* | | | | *inception* | | | | *WRN28-10* | | | |
| | CE | **65.09 ± 0.32** | 25.71 ± 0.28 | 73.52 ± 2.52 | 15.58 ± 0.27 | **64.30 ± 0.34** | **29.30 ± 0.27** | 69.05 ± 2.06 | 15.66 ± 0.44 | **68.25 ± 0.13** | 28.08 ± 0.43 | 68.30 ± 5.26 | 13.84 ± 0.66 |
| | LN | 64.90 ± 0.14 | **26.72 ± 0.19** | **78.67 ± 3.82** | 25.38 ± 0.34 | 63.87 ± 0.47 | 28.50 ± 0.58 | **76.99 ± 2.30** | 24.24 ± 1.08 | 67.84 ± 0.30 | **29.17 ± 0.22** | **79.88 ± 5.43** | 23.63 ± 1.10 |
| | DICE | 2.87 ± 2.81 | 1.59 ± 1.45 | 43.98 ± 8.98 | 21.60 ± 24.28 | 0.50 | 0.50 | 50.00 | 0.50 | 0.50 | 0.50 | 50.00 | 0.50 |
| | MM | 59.16 ± 0.22 | 20.43 ± 0.43 | 69.09 ± 3.15 | 22.99 ± 1.11 | 48.49 ± 0.41 | 18.17 ± 0.17 | 68.14 ± 2.69 | 19.05 ± 0.45 | 63.13 ± 0.37 | 23.19 ± 0.25 | 70.22 ± 3.38 | 23.83 ± 0.60 |
| | HEM- | 63.36 ± 0.23 | 25.47 ± 0.42 | 77.26 ± 5.66 | **31.42 ± 0.67** | 60.33 ± 0.73 | 26.37 ± 0.43 | 72.18 ± 1.13 | **26.84 ± 1.30** | 67.25 ± 0.52 | 28.09 ± 0.29 | 78.85 ± 2.55 | **28.53 ± 0.88** |
| **IN** | | *ResNet50* | | | | *SwinT* | | | | *ViT-B/16* | | | |
| | CE | **68.05 ± 0.10** | **30.61 ± 0.27** | 72.88 ± 4.22 | 12.22 ± 0.51 | 66.77 ± 0.28 | 24.53 ± 0.23 | 74.95 ± 7.63 | 11.87 ± 0.25 | 52.35 | 24.80 | 62.64 | 16.72 |
| | LN | 67.82 ± 0.25 | 30.13 ± 0.37 | **85.53 ± 1.02** | 21.12 ± 0.53 | **70.90 ± 0.03** | **28.01 ± 0.60** | 78.09 ± 5.15 | 9.80 ± 0.25 | **53.66** | **25.27** | 64.30 | 13.42 |
| | DICE | 1.67 ± 0.70 | 0.96 ± 0.39 | 43.83 ± 4.17 | **84.76 ± 7.59** | 16.18 ± 27.85 | 6.56 ± 9.19 | 51.34 ± 7.44 | 8.85 ± 8.23 | 5.98 | 3.71 | 38.17 | 11.72 |
| | MM | 55.75 ± 0.32 | 22.36 ± 0.22 | 60.50 ± 1.60 | 25.37 ± 0.49 | 24.57 ± 26.53 | 8.30 ± 8.51 | 54.16 ± 11.96 | 11.07 ± 9.52 | 46.89 | 21.00 | 56.75 | 20.63 |
| | HEM- | 67.10 ± 0.19 | 29.14 ± 0.39 | 81.77 ± 2.86 | 35.59 ± 1.09 | 66.92 ± 0.36 | 24.71 ± 1.00 | **85.09 ± 3.70** | **34.65 ± 0.78** | 49.83 | 22.81 | **67.12** | **25.06** |

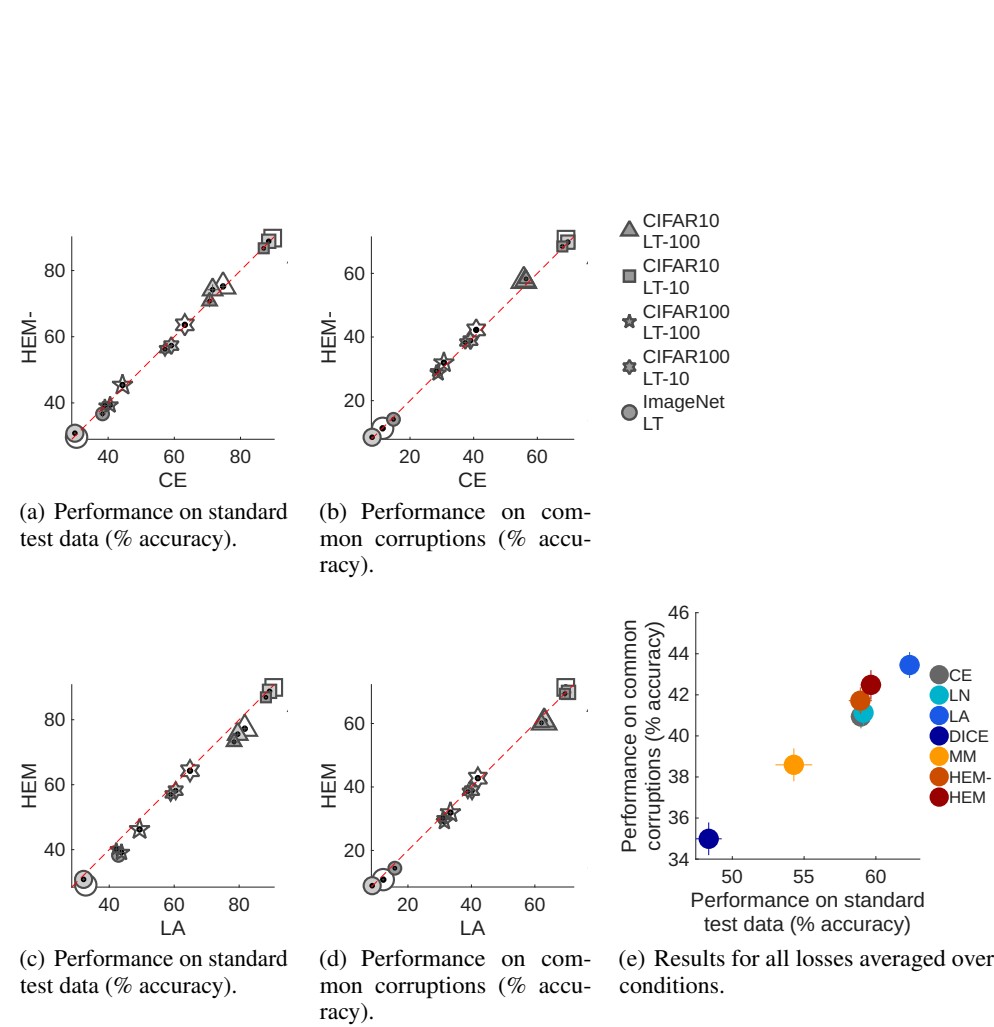

(a) Performance on standard test data (% accuracy).

(b) Performance on common corruptions (% accuracy).

(c) Performance on standard test data (% accuracy).

(d) Performance on common corruptions (% accuracy).

(e) Results for all losses averaged over conditions.

Figure 5: Results when learning with imbalanced data-sets and testing on clean and corrupt images. (a) and (b) directly compare the performance produced by HEM- and cross-entropy (CE) losses when used to train networks with long-tailed (LT) CIFAR10, CIFAR100, and ImageNet data-sets. Three different network architectures where used with each data-set, and the size of the marker used corresponds to the size of the network (sizes increase from left to right in Table 5). (c) and (d) show the same comparisons for HEM and LA losses. (e) Shows results averaged over all the data-sets and network architectures (and five trials in each condition) for all relevant losses: cross-entropy (CE), LogitNorm (LN), logit-adjusted (LA), DICE, multi-class margin (MM), HEM-, and HEM losses. The format of this figure is otherwise the same as, and described in the caption of, Fig. 3.

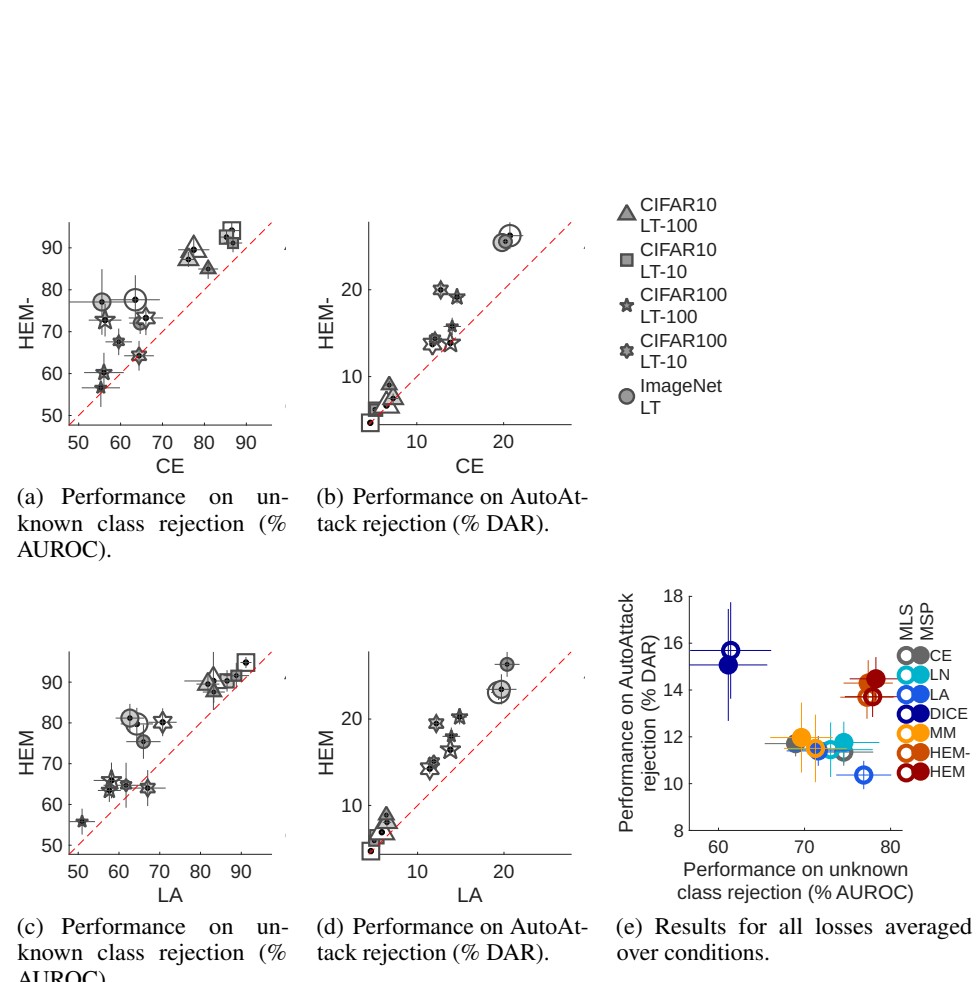

(a) Performance on unknown class rejection (% AUROC).

(b) Performance on AutoAttack rejection (% DAR).

(c) Performance on unknown class rejection (% AUROC).

(d) Performance on AutoAttack rejection (% DAR).

(e) Results for all losses averaged over conditions.

Figure 6: Results when learning with imbalanced data-sets and testing on unknown and adversarial images. This figure has an identical format to Fig. 5 except (a) and (c) compares performance of pairs of losses in terms of the ability to distinguish samples from known and unknown classes, and (b) and (d) compares the performance of pairs of losses in terms of the ability to deal correctly with adversarially perturbed samples. In (a) to (d) Maximum Softmax Probability (MSP) is used as the confidence score. (e) Shows results averaged over all the data-sets and network architecture (and five trials in each condition) for all relevant losses. Closed markers indicate that Maximum Softmax Probability (MSP) was used as the confidence score, while open markers plot results when using Maximum Logit Score (MLS). The format of this figure is otherwise the same as, and described in the caption of, Fig. 4.

Table 5: A comparison of the performance produced by different losses when applied to learning with imbalanced data-sets. Bold text indicates the best performance on each metric for each combination of training data-set and network architecture.

| Task / Loss | Clean Acc. (%) | Corrupt Acc. (%) | OOD AUROC (%) | AA DAR (%) | Clean Acc. (%) | Corrupt Acc. (%) | OOD AUROC (%) | AA DAR (%) |
|---|---|---|---|---|---|---|---|---|
| **CIFAR10LT-100** | *ResNet32* | | | | *ResNet18* | | | |
| CE | 70.65 ± 1.81 | 56.46 ± 0.57 | 80.89 ± 2.37 | 6.82 ± 0.47 | 74.73 ± 0.87 | 55.77 ± 1.26 | 77.45 ± 3.72 | 6.51 ± 0.22 |
| LN | 71.01 ± 2.08 | 54.61 ± 1.13 | 81.20 ± 2.60 | 7.02 ± 0.42 | 73.21 ± 0.42 | 54.18 ± 1.42 | 87.35 ± 2.82 | 6.59 ± 0.43 |
| LA | **78.44 ± 1.48** | **62.09 ± 0.84** | 83.25 ± 2.71 | 6.38 ± 0.33 | **81.72 ± 0.74** | **62.93 ± 0.66** | 83.16 ± 7.13 | 5.86 ± 0.11 |
| DICE | 62.81 ± 0.90 | 49.04 ± 1.12 | 58.44 ± 4.00 | **9.34 ± 1.50** | 70.32 ± 0.63 | 52.71 ± 0.89 | 73.71 ± 2.62 | **8.91 ± 0.52** |
| MM | 70.21 ± 1.55 | 53.18 ± 0.45 | 76.36 ± 3.07 | 7.19 ± 0.43 | 74.15 ± 0.58 | 56.36 ± 1.13 | 82.17 ± 1.45 | 7.17 ± 0.53 |
| HEM- | 70.73 ± 1.16 | 58.25 ± 1.28 | 84.95 ± 0.99 | 9.03 ± 0.70 | 75.21 ± 1.33 | 57.64 ± 0.43 | 89.54 ± 0.79 | 6.67 ± 0.48 |
| HEM | 73.19 ± 1.88 | 60.21 ± 1.43 | 87.57 ± 1.95 | 8.88 ± 0.90 | 77.24 ± 1.00 | 60.34 ± 1.10 | 90.29 ± 1.18 | 6.90 ± 0.39 |
| **CIFAR10LT-10** | *ResNet32* | | | | *WRN22-10* | | | |
| CE | 87.03 ± 0.57 | 67.86 ± 0.81 | 86.82 ± 2.19 | 5.18 ± 0.23 | 89.74 ± 0.38 | 69.02 ± 0.72 | 86.52 ± 1.89 | 4.64 ± 0.18 |
| LN | 86.81 ± 0.26 | 67.71 ± 0.78 | 91.96 ± 0.80 | 5.00 ± 0.62 | 89.16 ± 0.19 | 69.01 ± 0.56 | 94.64 ± 1.06 | 4.65 ± 0.35 |
| LA | 88.24 ± 0.27 | 69.47 ± 1.13 | 88.78 ± 3.13 | 4.97 ± 0.19 | **90.65 ± 0.13** | 69.73 ± 0.89 | 91.14 ± 1.35 | 4.59 ± 0.25 |
| DICE | 80.02 ± 0.37 | 60.33 ± 1.01 | 79.58 ± 2.80 | **7.53 ± 0.33** | 85.92 ± 0.16 | 64.81 ± 0.61 | 87.54 ± 1.33 | **5.75 ± 0.24** |
| MM | 86.61 ± 0.34 | 67.00 ± 0.61 | 85.68 ± 2.60 | 5.13 ± 0.16 | 89.40 ± 0.35 | 68.80 ± 0.89 | 89.50 ± 3.64 | 5.14 ± 0.12 |
| HEM- | 86.73 ± 0.50 | 68.45 ± 1.17 | 91.15 ± 2.26 | 6.22 ± 0.27 | 89.80 ± 0.48 | 70.74 ± 1.08 | 94.12 ± 1.30 | 4.67 ± 0.24 |
| HEM | 86.90 ± 0.55 | 69.36 ± 1.06 | 91.59 ± 1.46 | 5.91 ± 0.63 | 89.95 ± 0.35 | **71.37 ± 0.78** | **94.81 ± 0.80** | 4.72 ± 0.16 |
| **CIFAR100LT-100** | *ResNet32* | | | | *WRN22-10* | | | |
| CE | 39.02 ± 0.65 | 28.31 ± 0.51 | 55.29 ± 4.56 | 14.08 ± 1.01 | 44.35 ± 0.49 | 30.64 ± 0.36 | 56.30 ± 3.90 | 13.86 ± 0.36 |
| LN | 39.80 ± 0.40 | 27.36 ± 0.63 | 60.77 ± 5.10 | 10.65 ± 0.58 | 44.06 ± 0.39 | 30.68 ± 0.42 | 70.38 ± 6.79 | 14.61 ± 1.41 |
| LA | 42.16 ± 1.87 | 31.01 ± 0.94 | 50.95 ± 3.18 | 13.91 ± 1.03 | 49.34 ± 0.50 | 33.38 ± 0.56 | 58.16 ± 4.37 | 13.79 ± 0.37 |
| DICE | 32.98 ± 0.90 | 21.91 ± 0.25 | 48.94 ± 5.66 | 18.77 ± 0.44 | 41.29 ± 0.24 | 27.66 ± 0.29 | 55.04 ± 4.91 | **20.48 ± 0.84** |
| MM | 34.49 ± 0.69 | 23.30 ± 0.57 | 51.75 ± 4.27 | 14.77 ± 1.12 | 42.48 ± 1.66 | 29.05 ± 1.41 | 57.48 ± 5.19 | 18.32 ± 1.51 |
| HEM- | 39.20 ± 1.01 | 29.28 ± 0.94 | 56.60 ± 1.90 | 15.77 ± 1.35 | 45.39 ± 0.61 | 31.92 ± 0.58 | 72.75 ± 3.39 | 13.87 ± 0.52 |
| HEM | 40.18 ± 1.26 | 30.20 ± 1.29 | 55.78 ± 6.17 | 18.00 ± 0.84 | 46.24 ± 0.79 | 31.92 ± 0.34 | 65.89 ± 6.44 | 16.45 ± 0.81 |
| **CIFAR100LT-10** | *ResNet18* | | | | *WRN22-10* | | | |
| CE | 57.19 ± 0.44 | 37.37 ± 0.62 | 59.55 ± 3.17 | 12.09 ± 0.76 | 63.18 ± 0.28 | 40.80 ± 0.03 | 66.04 ± 4.09 | 11.81 ± 0.36 |
| LN | 57.44 ± 0.38 | 37.02 ± 0.27 | 65.26 ± 6.84 | 9.48 ± 0.33 | 62.45 ± 0.28 | 41.13 ± 0.40 | 71.12 ± 5.14 | 13.58 ± 0.78 |
| LA | 58.90 ± 0.72 | 38.88 ± 0.73 | 61.74 ± 5.51 | 11.88 ± 0.31 | 64.88 ± 0.46 | 42.04 ± 0.46 | 70.71 ± 3.42 | 11.41 ± 0.59 |
| DICE | 49.17 ± 0.61 | 31.49 ± 0.47 | 53.99 ± 6.72 | 14.44 ± 0.49 | 63.48 ± 0.52 | 40.42 ± 0.38 | 63.26 ± 4.97 | 12.76 ± 0.80 |
| MM | 51.81 ± 0.20 | 32.56 ± 0.38 | 61.66 ± 6.91 | 11.56 ± 1.08 | 62.04 ± 1.82 | 39.04 ± 1.28 | 73.30 ± 2.57 | 12.73 ± 0.50 |
| HEM- | 56.23 ± 1.23 | 38.27 ± 0.60 | 67.57 ± 6.93 | 14.36 ± 1.68 | 63.59 ± 0.61 | 42.24 ± 0.44 | 73.27 ± 3.07 | 13.74 ± 0.90 |
| HEM | 57.01 ± 0.62 | 38.51 ± 0.62 | 64.70 ± 3.82 | 15.09 ± 1.40 | 64.30 ± 0.28 | 42.77 ± 0.64 | 80.16 ± 5.66 | 14.25 ± 0.91 |
| **ImageNetLT** | *SwinT* | | | | *ConvNeXt-tiny* | | | |
| CE | 38.31 ± 1.04 | 14.80 ± 0.32 | 64.67 ± 2.58 | 20.21 ± 0.32 | 30.46 ± 0.63 | 11.40 ± 0.19 | 63.51 ± 5.86 | 20.72 ± 1.54 |
| LN | 36.31 ± 0.52 | 14.99 ± 0.17 | 75.73 ± 4.46 | 23.38 ± 1.23 | 33.93 ± 0.42 | 13.58 ± 0.53 | 64.52 ± 5.44 | 15.31 ± 0.67 |
| LA | 42.85 ± 0.37 | 15.87 ± 0.31 | 66.00 ± 4.20 | 20.36 ± 1.44 | 32.78 ± 0.68 | 12.23 ± 0.31 | 64.38 ± 3.95 | 19.38 ± 0.86 |
| DICE | 0.41 ± 0.11 | 0.35 ± 0.11 | 38.42 ± 9.76 | 60.50 ± 19.57 | 4.50 ± 1.24 | 2.72 ± 0.45 | 44.69 ± 6.62 | 14.53 ± 2.05 |
| MM | 17.21 ± 0.36 | 6.21 ± 0.04 | 55.81 ± 2.63 | 17.88 ± 1.86 | 22.32 ± 0.57 | 8.30 ± 0.20 | 65.91 ± 5.95 | 20.37 ± 1.96 |
| HEM- | 36.69 ± 0.70 | 14.13 ± 0.16 | 72.05 ± 3.28 | 25.56 ± 1.70 | 29.64 ± 0.65 | 11.31 ± 0.25 | 77.61 ± 4.30 | 26.23 ± 1.34 |
| HEM | 38.25 ± 0.51 | 14.40 ± 0.19 | 75.40 ± 2.06 | 26.33 ± 1.24 | 29.15 ± 0.67 | 10.80 ± 0.27 | 79.75 ± 2.52 | 23.07 ± 2.45 |

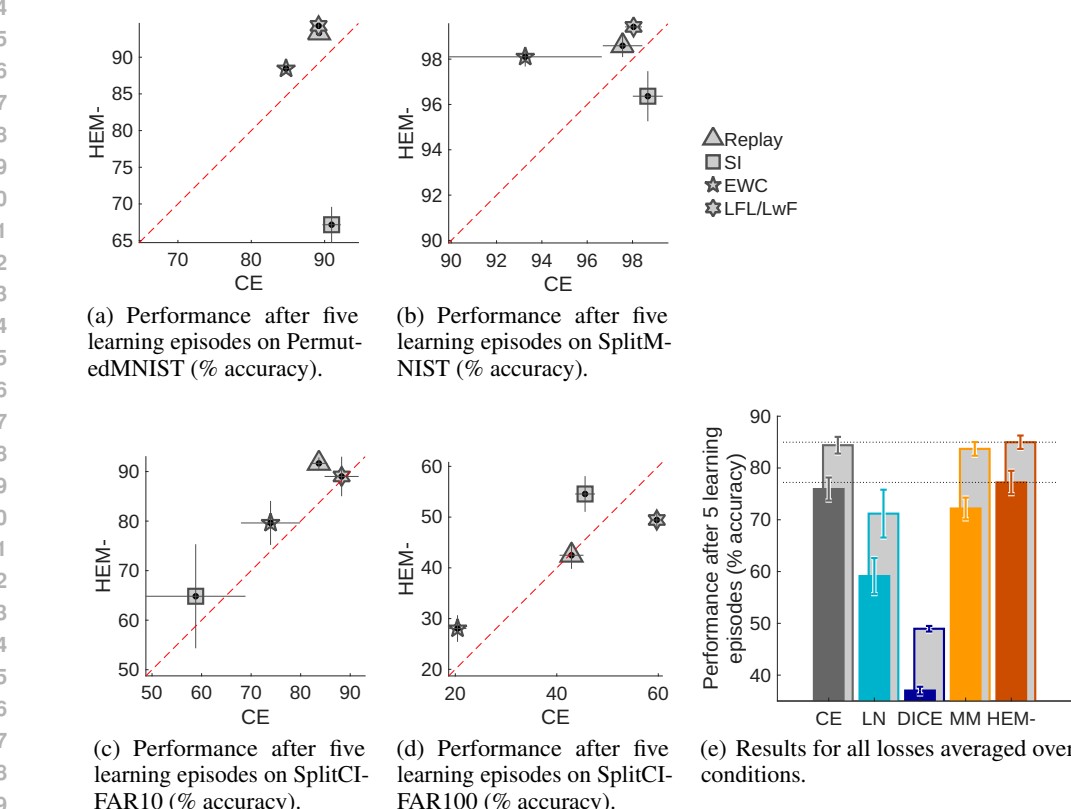

(a) Performance after five learning episodes on PermutedMNIST (% accuracy).

(b) Performance after five learning episodes on SplitMNIST (% accuracy).

(c) Performance after five learning episodes on SplitCIFAR10 (% accuracy).

(d) Performance after five learning episodes on SplitCIFAR100 (% accuracy).

(e) Results for all losses averaged over conditions.

Figure 7: Results for continual learning. (a) to (d) directly compare the performance produced by HEM and cross-entropy (CE) losses when applied to the PermutedMNIST, SplitMNIST, SplitCIFAR10 and SplitCIFAR100 tasks. Results above the diagonal are conditions where better performance was obtained when training with HEM rather than CE loss. Performance is measured as accuracy on the test data for all tasks after training on a sequence of five tasks. Error bars show the standard deviation recorded over five trials. Experiments were performed using a number of techniques to reduce the effects of catastrophic forgetting: Replay, Synaptic Intelligence (SI), Elastic Weight Consolidation (EWC), Less-Forgetful Learning (LFL), and Learning without Forgetting (LwF). LFL was used for PermutedMIST and LwF for the other tasks. (e) Shows results averaged over the four tasks and the four methods of reducing catastrophic forgetting applied to each task (and five trials in each condition) for all relevant losses: cross-entropy (CE), LogitNorm (LN), DICE, multi-class margin (MM), and HEM. Error bars show the mean standard deviation recorded across five trials in each condition. The light grey bars show results averaged over task and trials when for each loss only the best performing method of reducing catastrophic forgetting is chosen.

A comparison of results for all tested losses, averaged over the four conditions (and 5 trials per condition) can be seen in Fig. 6(e). The same results appear in the $7^{th}$ and $8^{th}$ segments of Fig. 1. In addition, Fig. 6(e) also shows performance when MLS is used as the rejection criterion. The numerical data can be found in the columns headed "OOD" and "AA" in Table 5.

## C.3 CONTINUAL LEARNING

Detailed results comparing the performance of CE and HEM trained networks for each individual condition together with error-bars can be seen in Figs. 7(a) to 7(d) for each of the four continual learning tasks. This is the data summarised in the $9^{th}$ segment of Fig. 2. A comparison of the performance of all tested losses averaged over tasks and conditions (and 5 trials per condition) are shown in Fig. 7(e). The same results appear in the $9^{th}$ segments of Fig. 1, but in terms of relative rather than absolute performance. The numerical data can be found in Table 6.

Table 6: A comparison of the performance produced by different losses when applied to continual learning. Bold text indicates the best performance for each combination of training data-set and method of reducing catastrophic interference.

| Task
Loss | Accuracy
(%) | Accuracy
(%) | Accuracy
(%) | Accuracy
(%) |
|---|---|---|---|---|
| PermutedMNIST | ___Reply___ | ___SI___ | ___EWC___ | ___LFL___ |
| CE | $89.23 \pm 0.34$ | $90.95 \pm 1.25$ | $84.74 \pm 0.33$ | $89.17 \pm 0.13$ |
| LN | $92.86 \pm 0.19$ | $83.64 \pm 1.61$ | $87.35 \pm 0.45$ | $\mathbf{94.92 \pm 0.23}$ |
| DICE | $37.61 \pm 2.01$ | $89.92 \pm 1.01$ | $9.74 \pm 1.44$ | $33.24 \pm 3.78$ |
| MM | $82.92 \pm 0.19$ | $\mathbf{92.95 \pm 0.20}$ | $75.03 \pm 0.99$ | $84.69 \pm 0.44$ |
| HEM- | $\mathbf{93.30 \pm 0.15}$ | $67.17 \pm 2.44$ | $\mathbf{88.45 \pm 0.48}$ | $94.28 \pm 0.33$ |
| SplitMNIST | ___Reply___ | ___SI___ | ___EWC___ | ___LwF___ |
| CE | $97.56 \pm 0.87$ | $98.68 \pm 0.66$ | $93.27 \pm 3.38$ | $98.05 \pm 0.37$ |
| LN | $63.86 \pm 3.77$ | $87.74 \pm 10.52$ | $53.65 \pm 6.02$ | $61.70 \pm 4.13$ |
| DICE | $50.17 \pm 0.67$ | $50.17 \pm 0.90$ | $50.53 \pm 0.93$ | $50.89 \pm 1.09$ |
| MM | $96.44 \pm 1.10$ | $\mathbf{98.76 \pm 0.68}$ | $80.31 \pm 3.73$ | $97.04 \pm 0.39$ |
| HEM- | $\mathbf{98.59 \pm 0.50}$ | $96.36 \pm 1.11$ | $\mathbf{98.10 \pm 0.43}$ | $\mathbf{99.42 \pm 0.14}$ |
| SplitCIFAR10 | ___Reply___ | ___SI___ | ___EWC___ | ___LwF___ |
| CE | $83.68 \pm 1.71$ | $58.81 \pm 10.06$ | $73.92 \pm 5.97$ | $88.270 \pm 3.444$ |
| LN | $69.66 \pm 5.89$ | $\mathbf{66.87 \pm 8.93}$ | $62.96 \pm 5.12$ | $69.59 \pm 4.90$ |
| DICE | $50.00 \pm 0.00$ | $50.00 \pm 0.00$ | $50.00 \pm 0.00$ | $50.00 \pm 0.00$ |
| MM | $81.32 \pm 3.34$ | $66.83 \pm 9.00$ | $74.31 \pm 3.83$ | $86.66 \pm 3.28$ |
| HEM- | $\mathbf{91.65 \pm 1.19}$ | $64.83 \pm 10.50$ | $\mathbf{79.63 \pm 4.44}$ | $89.01 \pm 3.99$ |
| SplitCIFAR100 | ___Reply___ | ___SI___ | ___EWC___ | ___LwF___ |
| CE | $\mathbf{42.90 \pm 2.36}$ | $45.57 \pm 1.95$ | $20.44 \pm 1.74$ | $\mathbf{59.70 \pm 1.09}$ |
| LN | $7.36 \pm 0.73$ | $32.42 \pm 1.82$ | $5.17 \pm 0.28$ | $6.27 \pm 1.07$ |
| DICE | $5.00 \pm 0.00$ | $5.00 \pm 0.00$ | $5.00 \pm 0.00$ | $5.00 \pm 0.00$ |
| MM | $30.12 \pm 2.09$ | $\mathbf{56.39 \pm 1.11}$ | $18.17 \pm 1.32$ | $32.84 \pm 1.89$ |
| HEM- | $42.48 \pm 2.68$ | $54.55 \pm 3.52$ | $\mathbf{28.04 \pm 2.62}$ | $49.43 \pm 1.61$ |

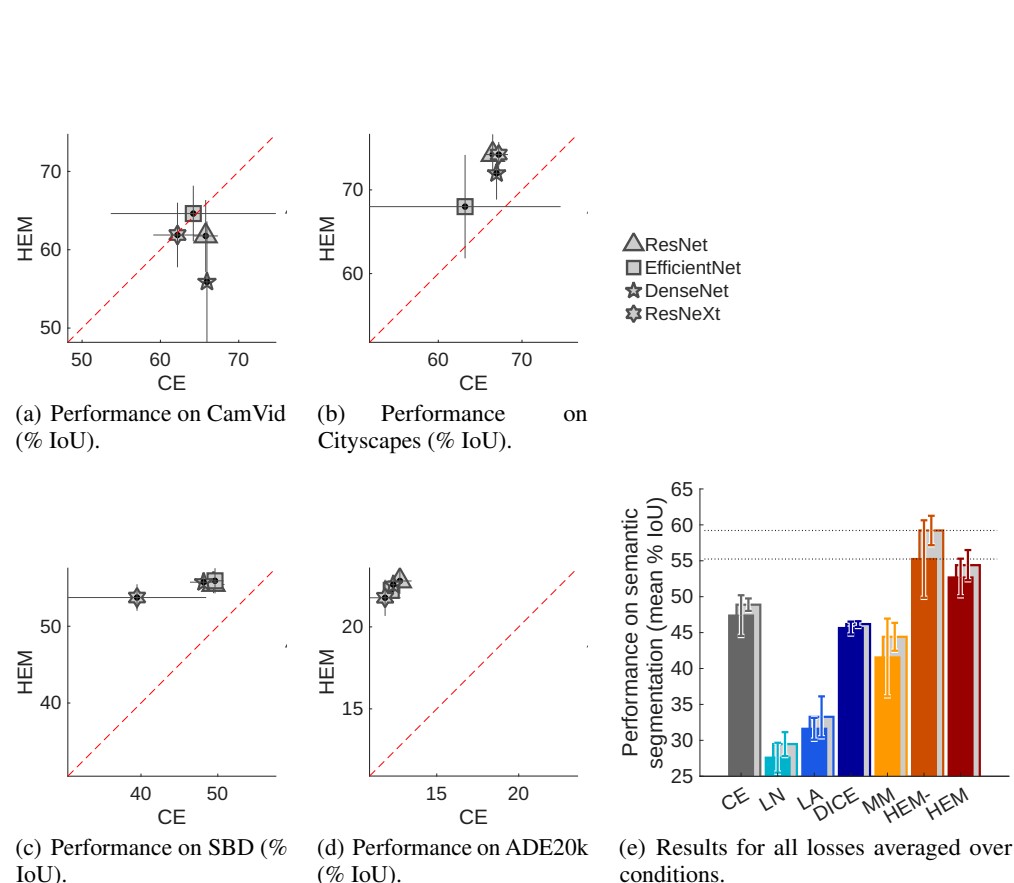

(a) Performance on CamVid (% IoU).

(b) Performance on Cityscapes (% IoU).

(c) Performance on SBD (% IoU).

(d) Performance on ADE20k (% IoU).

(e) Results for all losses averaged over conditions.

Figure 8: Results for semantic segmentation. (a), (b), (c) and (d) directly compare the performance produced by HEM and cross-entropy (CE) losses when applied to the CamVid, Cityscapes, SBD and ADE20k benchmarks. Results above the diagonal are conditions where better performance was obtained when training with HEM rather than CE loss. Performance is measured as mean percentage Intersection-over-Union (IoU). Results are averaged over five trials (four with ADE20k) performed for each encoder-backbone architecture (ResNet34, EfficientNet-b4, DenseNet201, and ResNeXt50). Error bars show the standard deviation recorded over these trials for each condition. (e) Shows results averaged over the four data-sets and four encoder-backbone architectures (and multiple trials in each condition) for all relevant losses: cross-entropy (CE), LogitNorm (LN), Logit-adjusted (LA), DICE, multi-class margin (MM), HEM-, and HEM. Error bars show the mean standard deviation recorded across trials in each condition. The light grey bars show results averaged over data-set and trials when for each loss only the backbone architecture that produces the best results is chosen.

Table 7: A comparison of the performance produced by different losses when applied to semantic segmentation. Bold text indicates the best performance for each combination of training data-set and and network architecture.

| Task | IoU | IoU | IoU | IoU |
|------|-----|-----|-----|-----|
| Loss | (%) | (%) | (%) | (%) |
| CamVid | _(ResNet34_ | _EfficientNet-b4_ | _DenseNet201_ | _ResNeXt50_ |
| CE | **65.79 ± 1.56** | 64.21 ± 10.52 | **65.92 ± 0.52** | **62.20 ± 3.07** |
| LN | 25.91 ± 2.94 | 27.67 ± 3.37 | 23.37 ± 2.42 | 20.43 ± 3.20 |
| LA | 39.61 ± 2.30 | 43.80 ± 1.30 | 40.72 ± 1.74 | 38.82 ± 1.86 |
| Focal | 65.78 ± 2.19 | 62.70 ± 10.08 | 63.80 ± 1.55 | 57.63 ± 9.20 |
| DICE | 57.79 ± 1.12 | 58.02 ± 0.82 | 58.17 ± 0.80 | 56.74 ± 1.74 |
| MM | 61.05 ± 3.03 | 61.51 ± 9.78 | 61.59 ± 1.28 | 53.07 ± 12.81 |
| HEM- | 58.97 ± 2.05 | 61.26 ± 1.64 | 60.97 ± 1.73 | 58.56 ± 2.89 |
| HEM | 61.77 ± 4.60 | **64.62 ± 3.55** | 55.89 ± 7.71 | 61.88 ± 4.11 |
| Cityscapes | _(ResNet34_ | _EfficientNet-b4_ | _DenseNet201_ | _ResNeXt50_ |
| CE | 66.49 ± 1.01 | 63.20 ± 11.43 | 66.98 ± 0.55 | 67.23 ± 1.11 |
| LN | 52.84 ± 1.55 | 46.19 ± 7.86 | 51.17 ± 1.41 | 49.38 ± 3.11 |
| LA | 54.63 ± 0.95 | 52.28 ± 0.80 | 55.84 ± 8.04 | 52.00 ± 0.27 |
| Focal | 68.27 ± 0.76 | 58.03 ± 14.14 | 68.16 ± 0.35 | 67.63 ± 0.78 |
| DICE | 70.44 ± 0.10 | 70.31 ± 1.81 | 70.97 ± 0.17 | 67.44 ± 6.27 |
| MM | 66.97 ± 13.21 | 60.81 ± 15.71 | 71.48 ± 1.82 | 65.83 ± 12.56 |
| HEM- | **80.02 ± 0.22** | **82.26 ± 0.61** | **81.28 ± 0.76** | **80.98 ± 0.19** |
| HEM | 74.25 ± 2.42 | 68.01 ± 6.20 | 71.98 ± 3.13 | 74.21 ± 1.52 |
| SBD | _(ResNet34_ | _EfficientNet-b4_ | _DenseNet201_ | _ResNeXt50_ |
| CE | 49.44 ± 1.41 | 49.64 ± 1.09 | 48.14 ± 1.76 | 39.51 ± 8.99 |
| LN | 19.83 ± 0.65 | 20.22 ± 2.37 | 20.35 ± 1.54 | 17.41 ± 1.05 |
| LA | 24.18 ± 0.87 | 27.11 ± 2.16 | 27.29 ± 1.71 | 24.08 ± 1.58 |
| Focal | 39.08 ± 8.50 | 48.63 ± 2.06 | 47.88 ± 2.67 | 44.38 ± 7.35 |
| DICE | 49.00 ± 0.00 | 49.00 ± 0.00 | 49.00 ± 0.00 | 49.00 ± 0.00 |
| MM | 33.74 ± 1.78 | 32.07 ± 1.60 | 36.02 ± 4.51 | 29.30 ± 5.02 |
| HEM- | 51.80 ± 5.03 | 29.59 ± 22.64 | 31.30 ± 20.84 | 37.96 ± 23.60 |
| HEM | **55.46 ± 1.14** | **55.91 ± 1.65** | **55.73 ± 0.96** | **53.77 ± 1.73** |
| ADE20k | _(ResNet34_ | _EfficientNet-b4_ | _DenseNet201_ | _ResNeXt50_ |
| CE | 12.76 ± 0.73 | 12.25 ± 0.47 | 12.34 ± 0.12 | 11.83 ± 0.93 |
| LN | 17.06 ± 0.19 | 16.70 ± 0.36 | 16.81 ± 0.58 | 16.00 ± 0.62 |
| LA | 6.16 ± 0.34 | 6.54 ± 0.13 | 6.48 ± 0.14 | 6.23 ± 0.23 |
| Focal | 12.12 ± 0.99 | 12.58 ± 0.08 | 12.50 ± 1.03 | 11.48 ± 0.45 |
| DICE | 6.60 ± 0.57 | 6.14 ± 0.24 | 6.93 ± 0.25 | 4.63 ± 0.33 |
| MM | 8.64 ± 0.13 | 9.35 ± 0.74 | 8.52 ± 0.51 | 4.74 ± 2.13 |
| HEM- | **41.54 ± 0.91** | **42.37 ± 1.61** | **42.99 ± 0.67** | **41.98 ± 0.87** |
| HEM | 22.82 ± 0.78 | 22.22 ± 0.74 | 22.58 ± 0.56 | 21.77 ± 1.09 |

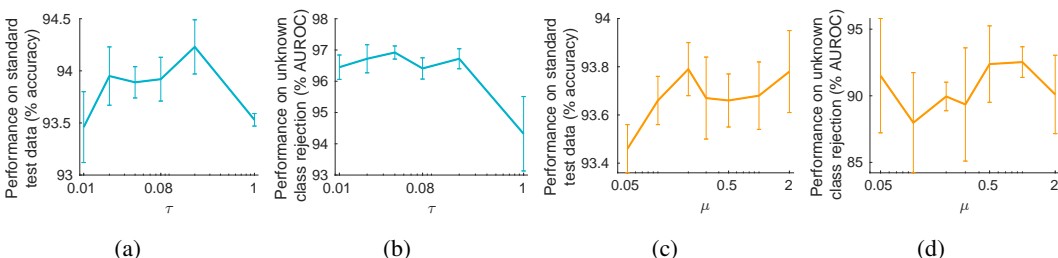

(a)  (b)  (c)  (d)

Figure 9: The effects of the loss hyper-parameter on LogitNorm (LN) and Multi-class Margin (MM) losses. Results for LN are shown in (a) and (b). Results for MM are shown in (c) and (d). Performance metrics are averaged over five trials performed with each parameter value, and the error bars show the standard deviation recorded across these five trials. Experiments were performed using the ResNet18 architecture trained using the standard training data for CIFAR10. Performance was evaluated using accuracy on the standard (clean) test data-set (a) and (c), and using AUROC to evaluate the accuracy with which known and unknown classes can be distinguished when Maximum Softmax Probability is used as the confidence score (b) and (d).

## C.4 SEMANTIC SEGMENTATION

Detailed results comparing the performance of CE and HEM trained networks for each backbone architecture together with error-bars can be seen in Figs. 8(a) to 8(d) for each of the four semantic segmentation benchmarks. This is the data summarised in the $10^{th}$ segment of Fig. 2. A comparison of the performance of all tested losses averaged over datasets and architectures (and multiple trials per condition) are shown in Fig. 8(e). The same results appear in the $10^{th}$ segments of Fig. 1, but in terms of relative rather than absolute performance. The numerical data can be found in Table 7. This table also includes results for Focal loss, a popular loss for segmentation tasks. Focal loss (Lin et al., 2017; Mukhoti et al., 2020) is a variant of CE loss that reduces the push towards infinite confidence. It defines a scaling factor that modifies the CE loss so that samples that are well classified (*i.e.*, have low CE loss) have even lower Focal loss. This hyper-parameter was set to a value of 2 in our experiments, which is the commonly used default value. Overall Focal loss performed similarly to CE loss (sometimes better, sometimes worse), and hence, much worse than HEM. The condition in which Focal loss out-performs CE loss by the largest margin was for the SBD data-set using the ResNeXt50 backbone. Here, CE achieves an IoU of 39.5% while Focal loss achieves 44.4%. However, this is still far behind HEM which achieves 53.8%.

## D SUPPLEMENTARY EXPERIMENTS

### D.1 LOSS HYPER-PARAMETER SELECTION

One of the great advantages of CE loss is that it does not introduce additional hyper-parameters that need to be tuned for different network architectures and tasks. Ideally, an alternative loss should also work without the need for hyper-parameter tuning. Preliminary experiments were performed to select an appropriate value for the hyper-parameter of each loss function that introduces such a parameter. These experiments were carried out using ResNet18 networks trained on CIFAR10: a combination of network architecture and data-set that was not used in the main experiments. The training set-up was as described in Appendix B.1 for training other ResNets, WRNs and PARN on CIFAR data. Results for these preliminary experiments are shown in Fig. 9 for LN and MM losses, and Fig. 10 for HEM loss.

Because the CIFAR10 training data is balanced, HEM is equivalent to HEM-, and we only consider a single, shared, margin $\mu$. For HEM, we expected that the results would be insensitive to the choice of $\mu$ as learning would scale the magnitude of the logits to match the chosen margin. Consistent with this expectation, the accuracy in classifying the CIFAR10 test set was fairly constant for networks trained using margin values ranging over more than two orders of magnitude (Fig. 10(a)). The choice of margin does, however, effect the ability to differentiate known and unknown classes using the

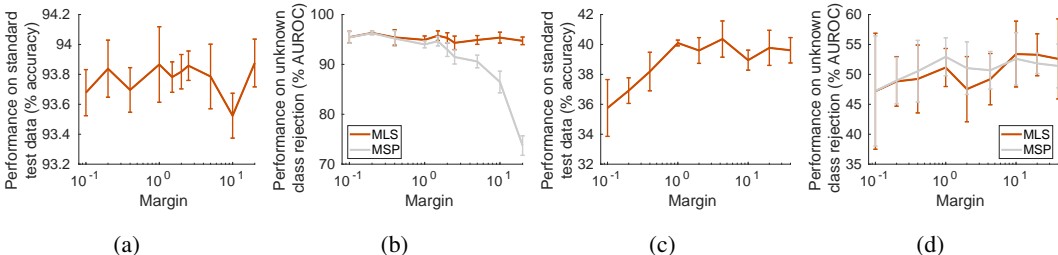

(a)              (b)              (c)              (d)

Figure 10: The effects of the HEM loss margin. Performance metrics are averaged over five trials performed with each margin value, and the error bars show the standard deviation recorded across these five trials. Experiments were performed using the ResNet18 architecture. Results in (a) and (b) are for networks trained using the full CIFAR10 training dataset. Results in (c) and (d) are for networks trained using a reduced CIFAR10 training dataset containing 50 samples per class. (a) and (c) show the effect of the margin on the accuracy of classifying the CIFAR10 test-set. (b) and (d) show the effects of the margin on the ability to identify, and reject, samples from unknown classes. Performance is averaged over seven out-of-distribution data-sets and the rejection criteria is based on either Maximum Softmax Probability (MSP) or Maximum Logit Score (MLS).

Maximum Softmax Probability (MSP) confidence score, as shown in Fig. 10(b). A large margin will cause the network to learn to produce high magnitude logits. The softmax function applied to larger magnitude logits will produce a more peaked distribution. As a result, the confidence in the prediction being made when measured using MSP, for both known and unknown classes, will be higher and it will become more difficult to distinguish known from unknown classes. However, even with a large margin, it is possible perform unknown class rejection if Maximum Logit Score (MLS) is used as the measure of prediction confidence (Fig. 10(b)).

If the the margin is reduced so that it approaches zero (or becomes negative) performance should degrade, as the classifier will not have learnt to produce higher logits for the correct class. For example, ResNet18 networks trained on CIFAR10 with HEM loss and $\mu = 0$ have mean standard test-set accuracy of 90.7% (*cf.*, with the results in Fig. 10(a)). We expected that the point at which the performance would degrade would depend on the number of training exemplars. When there are few training exemplars a larger margin is likely to be required in order allow accurate generalisation, whereas, when there are many training exemplars the decision boundary can be positioned more accurately and a smaller margin is sufficient to separate samples from different classes. To demonstrate this the previous experiments were repeated using a version of the CIFAR10 training data-set that contained only 50 samples per class (rather than the 5000 samples per class in the full CIFAR10 training set). As can be seen from Fig. 10(c), a larger margin is required to reach the upper limit of accuracy in this case. Based on these results it was decided to set the margin to be equal to $\sqrt{M/\sum_{i=1}^{n} s_i}$, where $s_i$ is the number of samples in class $i$ and $M$ was fixed at 2000. This equates to $\mu = 0.2$ for the full CIFAR10 training set, and $\mu = 2$ for the 50 samples per class version.

### D.2 ANALYSIS OF PREDICTION CONFIDENCE

As expected given the analysis in Section 2.1, CE loss tends to produce very high confidence for most samples (Fig. 11(a)). In contrast, the margin-based losses produce a much wider range of prediction confidence values for the known data (Figs. 11(b) and 11(c)). This was expected as none of these losses can be optimised by increasing the magnitude of the logits vector, and hence, the MSP. This confirms that the advantages in unknown class rejection we observe for HEM is indeed due to less severe overconfidence. MM loss fails to improve unknown class rejection performance beyond that of CE loss, and typically results in lower accuracy on the standard test data, particularly for data-sets with a large number of classes. As discussed in Section 2.2, an explanation for these empirical observations is that the MM loss tends to become close to zero prior to all samples being correctly classified (especially when $n$ is large), and hence, MM loss effectively terminates weight updates prematurely.

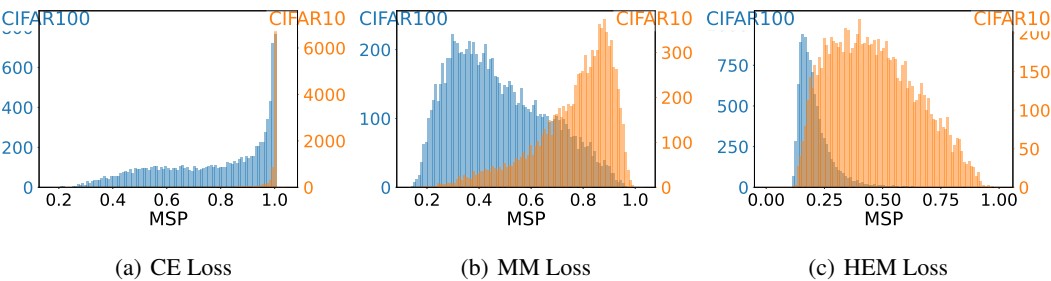

(a) CE Loss          (b) MM Loss          (c) HEM Loss

Figure 11: Prediction confidence after learning with standard data-sets. Results are for WRN22-10 networks trained on CIFAR10. Each graph shows histograms of the number of samples classified with different levels of prediction confidence (MSP). Separate histograms are shown for the response generated to unseen samples from known classes (the CIFAR10 test set) and unknown classes (the CIFAR100 test set). The former is measured against the right-hand vertical axis and the latter against the left-hand vertical axis.

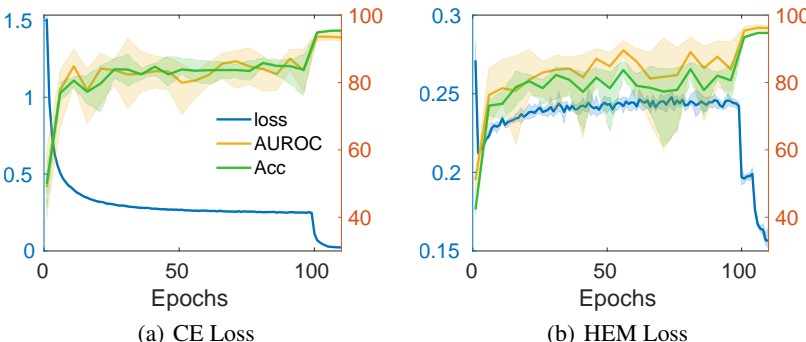

(a) CE Loss          (b) HEM Loss

Figure 12: Learning dynamics for WRN22-10 networks trained on CIFAR10 with (a) cross-entropy (CE) loss, (b) high error margin (HEM) loss. Each graph shows the change during training of the loss, the mean percentage AUROC averaged over seven data-sets containing unknown classes (see Appendix B.1.2), and the percentage clean accuracy on the standard test-set. The loss is measured against the left-hand vertical axis and the others two metrics against the right-hand axis. The solid lines show the mean values over five trials, and the shaded regions indicate the minimum and maximum values recorded in any of the five trials.

### D.3 ANALYSIS OF LEARNING

To check that our loss leads to equally effective learning as CE loss we investigated the changes in various metrics over the course of training (see Fig. 12). A major difference between CE and HEM loss is that the latter does not monotonically reduce over the whole course of training. This is to be expected, as only errors greater than the average contribute to the loss. Hence, it is possible that parameter updates during learning cause errors to move from just above the average to below the average. This will increase the mean of the remaining errors. Hence, it is important to prevent the calculation of the average from being used in the calculation of the gradients.

It can be seen that HEM benefits most from the drop in learning rate near the end of training and that there are large fluctuations in the loss, and the other recorded metrics, before the learning rate drop at 100 epochs. Both these observations suggest that HEM might benefit from a lower initial learning rate. This was confirmed experimentally by reducing the initial learning rate from 0.1 to 0.05. This increased performance for WRN22-10 networks trained on CIFAR10 with HEM loss on all the metrics used in this paper: the mean clean accuracy increased from 94.73% to 95.32%, the mean accuracy on common corruption increased from 75.08% to 75.41%, the mean AUROC for unknown class rejection increased from 96.16% to 96.56%, and mean DAR for adversarial attacks

Table 8: The effects of random oversampling, a complementary approach for dealing with training data imbalance, on the performance of CE and HEM- losses. Results are for the CIFAR10LT-100 data-set and the WRN22-10 architecture.

| Loss | Clean Acc. (%) | Corrupt Acc. (%) | OOD AUROC (%) | AA DAR (%) | Complementary Method | Clean Acc. (%) | Corrupt Acc. (%) | OOD AUROC (%) | AA DAR (%) |
|---|---|---|---|---|---|---|---|---|---|
| CE | $74.73 \pm 0.87$ | $55.77 \pm 1.26$ | $77.45 \pm 3.72$ | $6.51 \pm 0.22$ | oversampling | $73.67 \pm 0.86$ | $55.26 \pm 0.55$ | $80.86 \pm 1.84$ | $6.87 \pm 0.20$ |
| HEM- | $\mathbf{75.21 \pm 1.33}$ | $\mathbf{57.64 \pm 0.43}$ | $\mathbf{89.54 \pm 0.79}$ | $6.67 \pm 0.48$ | oversampling | $71.89 \pm 1.35$ | $55.83 \pm 1.57$ | $89.03 \pm 2.22$ | $\mathbf{8.21 \pm 0.59}$ |

Table 9: The effects of adversarial training, a complementary approach for improving adversarial robustness, on the performance of CE and HEM- losses. Results are for the CIFAR10 data-set and the WRN22-10 architecture. Adversarial training was performed using 10 steps of Projected Gradient Descent (PGD) and the maximum allowed perturbation was constrained by the $l_\infty$-norm to be less than $\frac{8}{255}$.

| Loss | Clean Acc. (%) | Corrupt Acc. (%) | OOD AUROC (%) | AA DAR (%) | Complementary Method | Clean Acc. (%) | Corrupt Acc. (%) | OOD AUROC (%) | AA DAR (%) |
|---|---|---|---|---|---|---|---|---|---|
| CE | $\mathbf{95.43 \pm 0.18}$ | $76.10 \pm 0.67$ | $93.31 \pm 1.01$ | $3.21 \pm 0.20$ | $\mathrm{PGD}_{l_\infty}^{10}$ | $88.04 \pm 0.21$ | $\mathbf{79.73 \pm 0.16}$ | $77.92 \pm 2.35$ | $68.72 \pm 0.42$ |
| HEM- | $94.73 \pm 0.06$ | $75.08 \pm 0.86$ | $\mathbf{96.16 \pm 0.48}$ | $5.91 \pm 0.38$ | $\mathrm{PGD}_{l_\infty}^{10}$ | $86.42 \pm 0.88$ | $78.40 \pm 0.78$ | $78.10 \pm 2.00$ | $\mathbf{70.00 \pm 0.90}$ |

increased from 5.91% to 6.34%. Further improvements in performance might be expected by more carefully tuning the learning hyper-parameters for each task.

# E    LOSSES COMBINED WITH COMPLEMENTARY APPROACHES

The article introduces a new loss. We have, therefore, focused on evaluating this new loss in comparison with alternative loss functions. For some of the criteria that we have used in our assessments there exist methods for improving performance that can be used in conjunction with any loss, including HEM. Comprehensively testing a loss with all of these complementary techniques would be a very large under-taking, and hence, we leave that for future work. Here, we report only a few preliminary experiments combining HEM with some well-known complementary approaches. These methods, like all the others we have used, such as those for reducing catastrophic forgetting, have been developed to work well with CE loss. As well as evaluating existing methods with HEM, future work might also develop new techniques specifically designed to work well with HEM.

## E.1    COMPLEMENTARY METHODS FOR DEALING WITH IMBALANCED TRAINING DATA

Random oversampling is a standard, baseline, method for training with imbalanced data (Branco et al., 2015). This method changes the relative frequency with which training samples are selected, so that samples from all classes appear equally often in the training batches. The result of using this method with CE and HEM- losses is shown in Table 8. It can be seem that oversampling is ineffective, resulting in poorer clean accuracy with both losses. This is likely due to the well-known issue of overfitting to the oversampled samples (Branco et al., 2015). For the other metrics and this particular combination of data-set and network architecture, HEM- outperforms CE both with and without oversampling.

## E.2    COMPLEMENTARY METHODS FOR ADVERSARIAL ROBUSTNESS

Adversarial training (AT) is a standard, and highly effective, defence against adversarial attack. It is a data-augmentation technique where training images are modified by adversarial perturbations. Augmenting the training images using multiple steps of Projected Gradient Descent (PGD; Madry et al., 2018) has become a standard method of AT against which all other methods of adversarial defence are benchmarked. The effects of using this form of adversarial training with CE and HEM- losses are shown in Table 9. It can be seem that AT has similar effects for both losses: trading-off clean accuracy and OOD rejection performance for increased adversarial robustness and a slight increase for corrupt accuracy.

Table 10: Results for alternative methods of unknown class rejection when used with CE and HEM-losses.

| Loss | Data-set | Architecture | OOD AUROC (%) | | | |
|------|----------|--------------|-----|-----|--------|-----|
| | | | MSP | MLS | Energy | GEN |
| CE | CIFAR10 | ResNet32 | $91.92 \pm 1.16$ | $94.27 \pm 0.83$ | $94.28 \pm 0.90$ | $94.42 \pm 0.78$ |
| HEM- | CIFAR10 | ResNet32 | $\mathbf{95.20 \pm 0.71}$ | $95.15 \pm 0.71$ | $93.97 \pm 0.84$ | $94.06 \pm 0.82$ |
| CE | TIN | ResNet18 | $73.52 \pm 2.52$ | $74.03 \pm 1.81$ | $74.06 \pm 1.69$ | $73.32 \pm 1.72$ |
| HEM- | TIN | ResNet18 | $79.49 \pm 3.15$ | $\mathbf{79.53 \pm 3.14}$ | $77.63 \pm 3.05$ | $77.68 \pm 3.04$ |

### E.3 ALTERNATIVE METHODS FOR UNKNOWN CLASS REJECTION

Many methods have been proposed for detecting samples that come from unknown classes (Yang et al., 2022; Zhu et al., 2024; Tajwar et al., 2021; Szyc et al., 2023; Vojir et al., 2023). Here we test four representitive post-hoc rejection methods: ones that can be used without re-training the classifier or modifying its architecture. MSP and MLS, results for which have already been presented in earlier sections, and two additional methods, Energy score (Liu et al., 2020) and Generalized Entropy score (GEN; Liu et al., 2023). Energy score defines prediction confidence as the negative logarithm of the denominator of the softmax function applied the network output layer. GEN defines confidence as being inversely proportional to the entropy of the class probability distribution produced by the classifier. As shown in Table 10, we found that all these methods produced very similar results. Furthermore, none of these methods enhanced the OOD rejection ability of CE-trained networks to be better than that of HEM-trained networks.

