# OpenReview forum: "A margin-based replacement for cross-entropy loss"
_ICLR.cc/2026/Conference — Submitted to ICLR 2026_

### Official Review · Reviewer_fSV9 · 2025-10-21

**Soundness:** 3
**Presentation:** 3
**Contribution:** 3
**Rating:** 6
**Confidence:** 3

**Summary:**

The authors propose a novel margin-based loss function, High Error Margin (HEM), which selectively averages over the examples in a batch that contribute the most to the loss.
This approach is intended to mitigate the issue of diminishing gradients commonly observed in margin-based losses toward the later stages of training. The paper presents empirical evidence demonstrating the advantages of the proposed method across several benchmarks.

**Strengths:**

The HEM loss is tested across multiple datasets, learning tasks, and architectures. Statistical significance is provided, showcasing the effectiveness.

Hyperparameters are selected in a clear and reproducible manner and optimized for the cross-entropy baseline rather than for HEM.
I value this highly, as overselling the observed effects with suitably chosen hyperparameters happens regularly.

**Weaknesses:**

The paper lacks a rigorous theoretical foundation for the claims that HEM mitigates catastrophic forgetting, yields more stable gradients than the Margin Maximization (MM) loss, or performs better on imbalanced datasets. The supporting arguments are largely heuristic, with no analytical results or relevant literature references provided to substantiate them.

**Questions:**

I have no questions.

---

> ### Author Response · Authors · 2025-11-20
>
> Weaknesses
>
> Margin-based losses have strong theoretical support and a clear connection to the objective of separating regions of feature-space occupied by different classes. Such losses were preferred over CE loss in the era of machine learning prior to deep learning, for example to train support vector machines. Such losses are therefore included as standard in many machine learning toolboxes, such as pytorch. However, they are seldom used as they are poor at training DNNs. There is no theoretical reason for the poor performance of MM loss with DNNs, it is a purely practical issue. Given that the failure of MM to train good DNNs is a practical issue, it makes sense to address it through an empirical study. Our study demonstrates for the first time that a margin-based loss can be used to train deep learning classifiers to be competitive with, or out-perform, those trained with CE loss. While our hypotheses as to why a margin loss should work well on continual learning and imbalanced datasets are intuitive, they are supported by our empirical results. Even if we had rigorous mathematical analyses, these would not guarantee the method would work in practice, as demonstrated by MM loss’s failure to scale to DNNs.

---

### Official Review · Reviewer_zrRR · 2025-10-23

**Soundness:** 3
**Presentation:** 3
**Contribution:** 2
**Rating:** 4
**Confidence:** 4

**Summary:**

Cross-entropy (CE) loss remains the standard for training deep neural networks in classification tasks. The authors introduce High Error Margin (HEM) loss, a general-purpose alternative that can outperform CE across diverse scenarios, including unknown class rejection, adversarial robustness, imbalanced data learning, continual learning, and semantic segmentation. Extensive evaluations on various architectures and benchmarks show that HEM performs on par with CE for balanced clean and corrupted image classification, while substantially improving performance in other settings. Compared to specialized task-specific losses - such as LogitNorm for open-set recognition, logit-adjusted loss for imbalance handling, and DICE for segmentation - HEM achieves more robust and general improvements, establishing itself as a versatile replacement for CE loss.

**Strengths:**

- The authors address an interesting topic.
- I like that, in addition to the standard classification, other aspects such as unbalanced data, adversarial robustness, continual learning, and semantic segmentation are also considered.
- The small toy example (Table 1) is useful for illustrating the differences between or problems with losses.
- The paper is easy to read and understand. In particular, I think section 2 is appropriate as motivation. I find Appendix A useful for explaining the various losses.
- The authors' idea doesn't have a lot of novelty, but it is intuitive and also shows good results.
- The distinction between HEM and HEM- makes a comparison with other loss functions fair.
- In semantic segmentation, the results are really good, with improvements almost every time.
- The presentation of the experiments in the main paper is pleasant.
- All details for reproduction are provided and the code should be made available.
- There are numerous experiments. The choice of datasets and evaluation metrics is appropriate.

**Weaknesses:**

- In Figure 1, I would add information about how many different datasets and networks were averaged.
- The main paper states that 18 networks and 18 datasets are used. I find this somewhat misleading, as 5 datasets and 3 networks are initially used for the classification experiments (and another 4 for semantic segmentation), and the other datasets only represent extensions for the respective task.
- For the attack experiments, the MSP or MLS is used for the thresholding. Entropy would also be interesting at this point, as it often performs better.
- The models used for classification are all rather dated. I understand why Restnet is included for comparison purposes, as it is simply standard practice, but I would also test newer models.
- The comparison takes place in the normal classification setting as well as in semantic segmentation, where HEM is only successful in the latter. I would rather compare it to other tasks such as OOD detection and robustness against attacks, by replacing the CE loss with HEM in existing methods used for these tasks. It is well known that normally trained networks are not very robust against all kinds of threats, and even though HEM brings slight improvements here, the results are not comparable to those achieved with specially trained/created methods.

**Questions:**

- The direct comparison is the classification task, and HEM is always worse than CE at standard setting (even with common corruptions). For unknown classes (OOD), attacks, and imbalanced data, there are other methods that are specifically desgined/trained for this purpose (i.e., it is not expected that using only CE loss will obtain good results). So here, the comparison is only against the simplest/most logical baseline, but not against other methods that were actually created for this purpose. Shouldn't the comparison be made against other methods here?
- "As a result training with HEM is faster than training with CE (for example, it reduces training time by approximately 10% for a ResNet18 trained for 200 epochs on TinyImageNet)" Has this behavior also been observed in other experiments, particularly in semantic segmentation? If so, I would highlight that in the paper.
- "It can be seen that HEM benefits most from the drop in learning rate near the end of training and that there are large fluctuations in the loss, and the other recorded metrics, before the learning rate drop at 100 epochs." That's an important point, so if you stop training too early, you might not see any improvement. Was this observed in all experiments? Any suggestions on how to address this?

---

> ### Author Response · Authors · 2025-11-20
>
> Weaknesses
> 1. We have modified the caption of Fig.1 to include this information.
> 2. Rather than getting into a discussion about what is and isn’t a different dataset, we have simply changed the text to say “many”.
> 3. We have tested some of our trained networks using an entropy-based method, GEN (Li et al, CVPR 2023), and another commonly used post-hoc rejection method, Energy score (Liu etal, NeurIPS 2020). In all the cases we assessed these alternative methods produced results very similar to those produced by MSP and MLS. Please see Table 10 in the revised manuscript for these additional results. Based on these results, the advantage of HEM trained networks for OOD detection is not dependent on the exact OOD detection criterion. We see no reason why other detection methods should be more helpful for CE than for HEM.
> 4. We have used a wide variety of networks, ranging in age from MLP to SwinT. However, if there are specific architectures that you think we should test, we would be pleased to do so. What newer architectures do you think we should test?
> 5. For OOD rejection we have compared to a strong baseline (LN loss), and as stated above have updated the manuscript to include additional results for complementary criteria for rejection (Energy & GEN). For adversarial robustness many state-of-the-art methods can be used in conjunction with any loss, as they are variations on adversarial training (AT) which is a data augmentation technique. To demonstrate that such methods are compatible with HEM loss we have performed standard AT as described in section E.2 of the revised manuscript. From the results in Table 9, it can be seen that AT has similar effects with both losses: trading-off clean accuracy and OOD rejection performance for increased adversarial robustness and a slight increase for corrupt accuracy. Note that the drop in clean accuracy caused by AT is much larger than the difference in clean accuracy between the two losses.
>
> Questions
>
> 1. We introduce a new loss here. It is most reasonable to test this new loss in comparison to alternative losses. Our claim is that it has advantages in comparison to other losses, not that it solves all problems of deep learning. Testing all additional methods that could be used to improve OOD rejection, adversarial robustness, and the treatment of imbalanced data is not feasible or necessary for this claim. Such extensive testing does not seem to be expected of other innovations in machine learning either. For example, a new activation function is compared to alternative activation functions using a few datasets and DNN architectures. It is not required that this new activation function be tested with many different losses, or in combination with a huge range of other techniques (such as methods for OOD rejection, adversarial robustness, and dealing with imbalanced data).
> 2. Unfortunately, we did not anticipate the loss to affect training time. We therefore did not record training time in many experiments and training has been performed on several different hardware configurations, meaning that many of the timings we do have are not comparable. For this reason, we have not included training time in the results tables.
> To answer your question, we repeated a semantic segmentation experiment to obtain comparable training time information. Specifically, we trained the resnet34 backbone on CityScapes. Using CE loss training time was 23563.0s. For HEM it was 22568.1s. We therefore also see a reduction in training time when HEM is used for semantic segmentation, but the reduction is more modest than for image classification (4% rather than 10%). We have added this additional timing information to the relevant part of the conclusions.
> 3. For the particular example shown, stopping training early would reduce the performance obtained with HEM loss. However, the same would also be true for CE loss. The performance produced by any loss is highly dependent on the training recipe used. In order to perform controlled experiments where we can isolate the effects to a single change in set-up, we have used the same training recipe for both CE loss and HEM loss. This training recipe is based on ones that have been optimised through the work of many labs over many years to produce excellent performance with CE loss. HEM is therefore at a significant disadvantage in all the comparisons we have made. Despite this, HEM loss performs better than CE loss in 8 out of the 10 comparisons we have made, as shown if Fig.2.

---

> > ### Comment · Reviewer_zrRR · 2025-11-27
> >
> > Thanks for the response. After the explanations and additional experiments (also for the other reviewers), I increased my score from 4 to 6.

---

### Official Review · Reviewer_fZDa · 2025-10-31

**Soundness:** 3
**Presentation:** 3
**Contribution:** 2
**Rating:** 4
**Confidence:** 4

**Summary:**

The authors propose a loss function called “High Error Margin (HEM)” that should outperform cross-entropy (CE) in various classification tasks and metrics. It’s a margin-based loss, computed only over class errors higher than the mean, calculated for each data point. This makes the loss focus only on high-error logits. Moreover, the marginal loss seems beneficial because it assigns zero loss to already well-classified instances - addressing the main problem of CE, which continues penalizing samples that are already correctly classified, often leading to overfitting. The value proposition of this paper is clear but too broad - authors are trying to cover and improve too many classification applications (imbalanced datasets, continual learning, OOD detection, corrupted data accuracy, segmentation, and more). It’s impossible to really understand whether the proposed loss works without reading the appendices, which I see as a major flaw. The pitfalls of CE are well known, and while this HEM loss generally outperforms CE, in presented results, there’s always another loss that performs better than HEM in each application. Moreover, authors only tested a few losses considering the many tasks and metrics they aim to address.

**Strengths:**

The paper proposes the High Error Margin (HEM) loss, a margin-based alternative to cross- entropy (CE) for classification tasks. The motivation is clear: CE has well-documented limitations, including non-zero penalties for correctly classified samples and a tendency toward overconfident predictions on unseen data, leading to mis-calibration, poor robustness to out-of-distribution detection, and catastrophic forgetting.

HEM addresses these issues by adaptively averaging high-error logits (those whose errors exceed the mean across classes), thereby emphasizing misclassified or uncertain examples. The idea is in- tuitively appealing and well-motivated, as it seeks a general-purpose loss function that can perform more robustly than CE across many applications. The empirical evaluation is particularly comprehensive - covering diverse architectures, multiple datasets, and several application domains. The paper’s presentation is generally clear, and the problem statement is timely and relevant to ongoing research in loss function design and optimization.

**Weaknesses:**

The claim of general superiority is somewhat too broad. Figures 1 and 2 report a large number of experiments, but the main text provides insufficient explanation of these results and their configurations, with many key details relegated to the appendices.

Given the wide range of applications tested, comparing only three alternative losses (apart from CE and MM losses) may be insufficient.

For imbalanced data, in the literature CE is often used in combination with other techniques—such as random oversampling or sample reweighting—which are not considered here.

As also noted in the conclusion (lines 461–474), for each application there exists at least one specialized loss that outperforms HEM, except for semantic segmentation. However, for this specific task, the authors did not evaluate Focal Loss, another modification of CE that directly addresses the over-penalization of already well-classified samples.

Is it useful to propose a loss function that performs generally well, but falls short compared to specialized losses in each specific application?

**Questions:**

1. Please consider moving Appendix A.5 into the main body. It repeats Equation (1) as Equation (8), plus the MM formulation, which is only the average of the errors.
2. In the semantic segmentation experiments, Focal Loss should be included for comparison, since it is also a modification of CE that mitigates the issue of over-penalizing already well-classified samples.

---

> ### Author Response · Authors · 2025-11-20
>
> Weaknesses
> 1. Unfortunately, given the strict page limit and the large number of experiments we have performed, it is impossible to report all the results in the main body of the paper. If there are particular details that you think we need to provide in the main text, please let us know and we will modify the manuscript accordingly.
> 2. This already makes 5 losses that we compare to: CE (the most commonly used loss), LN (which is a strong baseline for unknown class rejection), LA (a strong baseline for training with imbalanced data), DICE (a strong baseline for semantic segmentation), and MM (the failing loss we improve upon). Additionally, we now test Focal loss for semantic segmentation as you suggested in your questions. We do not know of any important competing loss that we ignore. If you know any that you think would be important to compare to, please let us know.
> 3. For training with imbalanced data, LA loss produces far stronger performance than techniques such as random oversampling or sample reweighting. Given also that LA is a loss-based solution, we compare our loss primarily with LA. The techniques to deal with imbalanced data you suggest can be used in conjunction with any loss, including HEM.
> Nonetheless, we wanted to test whether additional techniques like random oversampling fundamentally change our results. To do so, we tested random oversampling with both CE and HEM- on one of the variants for the imbalanced CIFAR 10. The results are:
> | Loss | Clean Acc (%) | Corrupt Acc (%) | OOD AUROC (%) | AA DAR (%) |
> | :--- | :--- | :--- | :--- | :--- |
> | CIFAR10LT-100 WRN22-10 |  |  |  |  |
> | CE | 74.73 | 55.77 | 77.45  | 6.51 |
> | CE + random oversampling | 73.67 | 55.26 | 80.86 | 6.87 |
> | HEM- | 75.21 | 57.64 | 89.54 | 6.67 |
> | HEM- + random oversampling | 71.89 | 55.83 | 89.03 | 8.21 |
> Generally, oversampling was ineffective, resulting in poorer clean accuracy with both losses. This is likely due to the well-known issue of overfitting to the oversampled samples. For the other metrics HEM- outperforms CE both with and without oversampling. We have added these results to a new section of the manuscript (Sec E) describing preliminary results of combining HEM loss with complementary techniques that you and the other reviewers suggested.
> 4. There appears to be a slight misunderstanding here: HEM outperforms all other losses on two tasks: semantic segmentation and continual learning. It also outperforms all other losses in terms of robustness to unknown samples. If you consider the 10 segments of Fig.1 you can see that HEM is best in 5 out of the 10 evaluations we have performed. In contrast, CE loss is best in only 2! We have revised the conclusions to make sure that the point that HEM does perform best for quite a few tasks comes across.
> To respond to your concern about Focal loss, we have repeated the experiments on semantic segmentation using Focal loss, and have added these results to Table 7 of the manuscript. Overall Focal loss performed similarly to CE loss (sometimes better, sometimes worse), and hence, much worse than HEM. The condition in which focal loss out-performs CE loss by the largest margin is for the SBD data-set using the ResNeXt50 backbone. Here, CE achieves an IoU of 39.5% while Focal loss achieves 44.4%. However, this is still far behind HEM which achieves 53.8%.
> 5. HEM is the best loss for some tasks in our evaluations. Thus, if specialised losses are of interest, so is HEM as it can be considered a specialised loss for some applications. Furthermore, CE loss also fails to outperform other losses in some specific circumstances. Hence, your argument, that a loss that is not always superior is of no interest, would apply also to CE loss. In fact, it would apply more to CE loss than to HEM, as CE loss is only superior in 2 circumstances whereas HEM is superior in 5.
> Furthermore, we do believe having a loss that works well in most situations is desirable as new tasks come up regularly and having to search for a task specific best loss for each new application would be highly labor intensive. The fact that CE loss is used for the vast majority of applications is a testimony that engineers in many situations tend to prefer a loss that works well enough everywhere over using different specialized losses.
>
> Questions
>
> 1. We have considered describing the original MM loss in the main paper, but feel that it makes more sense to describe all existing losses together, and we do not have the space to move the whole of section A into the main body. Furthermore, it should be noted that Equations 1 and 8 differ: HEM allows different margins for each logit, while MM does not.
> 2. As discussed above, Focal loss does not perform better than HEM at semantic segmentation. We have updated the results in the manuscript to show this.

---

### Official Review · Reviewer_HT2C · 2025-11-04

**Soundness:** 2
**Presentation:** 3
**Contribution:** 3
**Rating:** 6
**Confidence:** 3

**Summary:**

This paper proposes High Error Margin (HEM) loss as a general-purpose alternative to cross-entropy (CE) for training deep neural networks. Through extensive experiments across multiple architectures and datasets, the authors claim HEM outperforms CE on five diverse tasks: unknown class rejection, adversarial robustness, imbalanced learning, continual learning, and semantic segmentation. HEM only underperforms CE on clean/balanced classification by a small margin. Notably, HEM also outperforms task-specific specialized losses (LogitNorm, Logit-adjusted, DICE) on most tasks, positioning it as a universal replacement for CE loss.

**Strengths:**

- The paper is well written.
- The proposed approach is technically sound.
- The empirical experiment conducted span a wide range of datasets.
- The proposed method seem to work on a wide range of problems.

**Weaknesses:**

- The proposed method seems to lack some theoretical justification. Some theoretical analysis on the proposed loss function, and on why the proposed HEM loss is better than regular margin loss can further strengthen the paper.
- The claim to replace CCE loss is somewhat aggressive to me. From figure 2, the proposed loss function still underperforms CCE loss in the clean-data scenarios pretty significantly.
- Following up on the previous point, in order to claim HEM as a "replacement" for CCE loss, additional experiments are needed in my opinion.
- Unlike CCE loss, the proposed loss function has a hyper-parameter that needs to be tuned.

**Questions:**

- Does the proposed loss function work in other scenarios like NLP, and with other tasks like language model pre-training, which also uses CCE?

---

> ### Author Response · Authors · 2025-11-20
>
> Weaknesses
>
> 1. Margin-based losses have strong theoretical support and a clear connection to the objective of separating the regions of feature-space occupied by different classes. Such losses predate deep learning as they were used to train support vector machines, for example. Margin-based losses are therefore included as standard in many machine learning toolboxes, such as pytorch. However, they are seldom used as they are poor at training DNNs. There is no theoretical reason for the poor performance of MM loss with DNNs, it is a purely practical issue. Given that the failure of MM to train good DNNs is a practical issue, it makes sense that our solution is evaluated practically, too. Our empirical study demonstrates for the first time that a margin-based loss can be used to train deep learning classifiers to be competitive with, or out-perform, those trained with CE loss.
> 2. The main claim we make is that HEM can be used as a replacement for CE loss and that it can be used equally broadly as CE loss. We acknowledge that HEM reduces clean accuracy in the paper already. In exchange for this decrease, HEM yields a large increase in performance of out-of-distribution data, while also improving performance on imbalanced data and semantic segmentation. We thus believe HEM is usually a good replacement, but we do not (want to) claim that CE loss should never be used, of course.
> 3. We have already evaluated HEM across a large range of tasks, datasets, and architectures. If there are specific experiments that we could do to convince the reviewer and that can reasonably fit into a short conference paper, we would be pleased to perform them. Are there any specific experiments you think are missing?
> 4. The margin is indeed a hyperparameter of the HEM loss, but performance of the loss appears to be largely unaffected by this parameter as we discuss in Appendix D1. In a few control experiments, HEM’s performance was fairly constant for networks trained using margin values ranging over more than two orders of magnitude. Furthermore, we have not found it necessary to tune M for the whole range of tasks, datasets, and architectures we tested for the main text.
> There are strong theoretical arguments for the lack of sensitivity of HEM to the choice of margin as well. Specifically, all network architectures we use here and almost all networks used in general can produce logits at any scale by rescaling weights in final readout layers. Thus, learning can simply scale the magnitude of the network’s outputs so that it matches any chosen margin. For example, any network trained with a margin m can be transformed into a good solution for a margin of 10m by multiplying the weights of the last linear layer by 10.
> This is analogous to CE loss, which also has a hyper-parameter that is often omitted: the temperature used for the softmax applied to the logits. A change of the temperature is one of the modifications made to CE to produce LogitNorm loss and the temperature does change the results produced. However, generally, CE loss works well without the need to tune the softmax temperature. Our results show that HEM also works well without the need to tune its hyper-parameter.
>
> Questions
>
> We do not see any reason why HEM should not work in places where CE loss is used for NLP like next token prediction as you propose. We do not want to make strong claims about this without a thorough test of HEM in this situation though. A thorough evaluation of HEM in other domains would require at least another paper’s worth of results and discussion. We therefore leave that for future work. We have modified the abstract to make it clearer that the scope of the current work is limited to the vision domain.

---

### Meta-Review · Area_Chair_9yHr · 2026-01-13

**Summary:**

This paper presents a simple new margin-based loss as a drop-in replacement to cross-entropy. Reviewers had a number of important and shared concerns, including1) The lack of theoretical justification or analysis on understanding why the method works, 2) Overclaims and concerns about the method significantly underperforming CE on clean data scenarios, 3) need for an additional hyper-parameter, and 4) need for additional experimentation such as more modern architectures.

**Reviewer Concerns:**

Concerns about the hyper-parameter are somewhat resolved, arguing its insensitivity (and, as mentioned, temperature-based CE also has one). Unfortunately, many of the other above concerns were not well-resolved. The proposed method is indeed empirical (which is not out of the ordinary in this field), but the empirical results were somewhat under-described in the original and also indeed significantly sacrifices clean performance, and if specialized settings are of interest then specialized methods in some cases also outperform the method (though not in all). As a result, it is still difficult to understand under what circumstances this method would be preferable, as real-world deployment would be significantly interested in the clean performance.

**Reviewer Scores:**

Based on the discussion above, it is unlikely in my assessment that the negative or positive reviewers (the latter of which also had the concerns above) would increase their scores. As a result, this paper is overall borderline leaning towards rejection at this time.

---

### Decision · Program_Chairs · 2026-01-26

Reject